



# Diffuse sources of TFA: atmospheric and terrestrial inputs, retention and pathways at the catchment scale

Immanuel Frenzel[1], Dario Nöltge[2], Finnian Freeling[3], Michael Müller[2], Jens Lange[1]

[1]Chair of Hydrology, University Freiburg i.Br., 79098 Freiburg, Germany
[2]Institute of Pharmaceutical Sciences, University Freiburg i.Br., 79104 Freiburg, Germany
[3]Technologiezentrum Wasser, 76139 Karlsruhe, Germany

*Correspondence to*: Immanuel Frenzel (immanuel.frenzel@hydrology.uni-freiburg.de)

## Abstract

Trifluoroacetate (TFA) is a contaminant from various human sources. The degradation of fluorinated gases in the atmosphere leads to a ubiquitous input through precipitation. Degradation of certain agricultural pesticides and wastewater-borne pharmaceuticals adds to the amount of TFA pollution. Once released into the aquatic environment, TFA is nearly conservative due to its negative charge, high water solubility, and absence of degradation pathways. Consequently, TFA concentrations in the environment are constantly increasing, following the production of precursor substances. Previous studies suggested the accumulation of TFA in plants and its retention in organic soil. This knowledge, however, is based on a small number of environmental samples or laboratory experiments. Catchment-scale studies are so far missing. In particular, hydrological processes controlling the retention and mobilization of TFA are poorly understood. Therefore, we analyzed a two-year dataset of weekly water samples for major ions and isotope tracers with TFA in the mountainous Dreisam catchment (Black Forest, Germany). We sampled precipitation, the discharge of three nested catchments, and a hillslope spring. A balancing approach suggested that TFA was not permanently retained in forested headwaters. Therefore, we were able to estimate evapotranspiration in the sub-catchments from two years of TFA concentrations in streamflow. In agricultural areas, we found a surplus of TFA, which totaled an annual input of $11.4 \pm 3.9$ kg km⁻² for arable land. A correlation analysis using environmental tracers, combined with knowledge of runoff generation in the study catchment, suggested that previously retained TFA was flushed from soils under wet conditions, with subsurface stormflow serving as a primary transport path. These findings indicate that TFA concentrations in soils may be higher than average concentrations found in rain or streamflow. Therefore, future research should focus on TFA retention in the unsaturated zone.

## 1. Introduction

Trifluoroacetate (TFA) is a degradation product of various anthropogenic fluorinated compounds. Ongoing emissions of fluorinated refrigerants have resulted in a ubiquitous input of TFA through precipitation (Franklin, 1993; Jordan and Frank,





1999; Berg et al., 2000; Freeling et al., 2020). In addition to atmospheric input, increased use of fluorinated plant protection products (PPPs) poses a terrestrial diffuse source of TFA (Scheurer et al., 2017; Arena et al., 2017; Bhat et al., 2022). Furthermore, point sources such as industrial sites or municipal wastewater treatment plants (WWTP) contribute to the overall TFA pollution. Once released, TFA is persistent and highly mobile in the aqueous phase (Boutonnet et al., 1999). Its stability raises concerns about the potential for accumulation (Arp et al., 2024). Elevated TFA concentrations have been documented

across various environmental media. For instance, Freeling et al. (2022) reported increasing TFA levels in the leaves and needles of four tree species over the last three decades in Germany, while Li et al. (2010) observed elevated TFA levels in soils. Notably, elevated and increasing concentrations of TFA have also been found in groundwater (Liang et al., 2023; Albers and Sültenfuss, 2024) and surface waters (Cahill et al., 2001; German Environment Agency, 2021; Cahill, 2022; Freeling and Björnsdotter, 2023). However, knowledge about TFA transport within environmental compartments remains limited. Two

field studies investigated TFA retention and transport over a half-year study period (Berger et al., 1997; Likens et al., 1997). Forests retained TFA in soils (with retention rates between 10–20% and 5–30%) and in plants (5–20% and 5–35%), whereas retention observed in wetlands was significantly higher, ranging from 20–60% for soils and 20–50% for plants. This discrepancy suggests considerable variability in TFA uptake and retention between different ecosystems. Moreover, both studies were situated in the same area, which limits transferability. Furthermore, they relied on short measurement periods

(half a year), and may have failed to capture longer-term processes.

Thus, we identify a general lack of comprehensive field studies concerning the accumulation, retention, and transport of atmospherically deposited TFA at the catchment scale. This gap extends to terrestrial TFA input from agriculture, since accurate data on the quantities of applied TFA-precursor PPPs on the catchment scale are usually not available. Consequently, recent meta-analyses relied on sales data and estimated application masses to derive potential TFA contributions from these

PPPs for European countries and the contiguous USA (German Environment Agency, 2023; Joerss et al., 2024). Moreover, the lack of transformation rates further complicates the quantification of released TFA amounts.

To address the fate and transport of TFA from terrestrial and atmospheric sources, we created a two-year dataset of TFA concentrations and loads in different environmental compartments. We chose the mesoscale Dreisam river catchment (DRC) as our study area, also because its hydrological processes have an extensive history of research (Hoeg et al., 2000; Hangen et

al., 2001; Uhlenbrook, 1999; Seibert et al., 2000; Uhlenbrook et al., 2002; Uhlenbrook and Leibundgut, 2002; Wenninger et al., 2004; Lange and Haensler, 2012). For two consecutive years, we took weekly samples of precipitation, in the Dreisam river at the catchment outlet, in two nested headwaters, and in a hillslope spring.

First, we correlated TFA dynamics in river and spring discharge with major water ions and stable water isotopes. Together with existing knowledge on runoff generation processes, we gained insights into the transport processes of TFA. Second, we

applied a balancing approach in the headwaters, free of arable land. This enabled us to determine the amount of retained TFA in natural catchments. Third, we assessed the TFA balance in the main catchment with mixed land use, acquiring insights into the surplus originating from agricultural sources. Finally, we elucidated the potential of catchment-scale TFA data to estimate actual evapotranspiration (*ET*). We aimed to answer the following research questions:





- Do atmospheric TFA inputs accumulate in catchments devoid of arable land?

- Are terrestrial TFA inputs from agriculture relevant?

- Which pathways dominate TFA transport?

- How reliable are *ET* estimates from TFA?



## 2. Materials and Methods

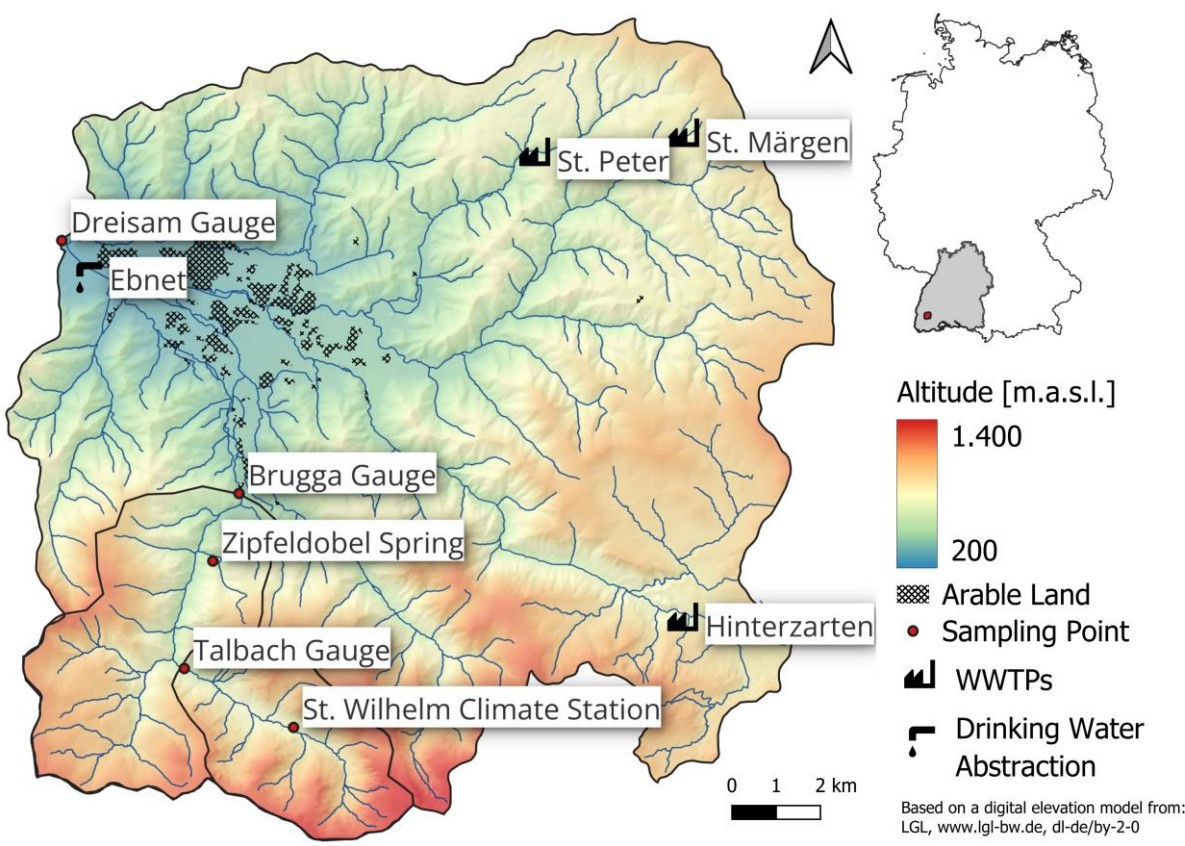

**Figure 1: The Dreisam catchment: sampling locations and sites with potential TFA input.**

### 2.1 The Dreisam river catchment (DRC)

The 257 km² DRC (Fig. 1) is located in the southern Black Forest, Baden-Württemberg, Southwest Germany. The mean annual air temperature ranges from 4.5 °C at the mountain peak of the Feldberg (1493 m a.s.l.) to 10.7 °C at the catchment outlet (309 m a.s.l.) in Ebnet, located in the eastern part of the city of Freiburg. Annual precipitation increases from 955 mm in the city of Freiburg to 1900 mm at the Feldberg (Fuchs et al., 2001). Bedrock consists of crystalline rocks, mainly granite and gneiss, overlain by unconsolidated glacial and periglacial drift covers of varying thickness. This loose material has led to the development of cambisols, with bedrock outcrops appearing at various locations (Tilch et al., 2002). The DRC comprises two distinct topographical units: (i) The lower part is characterized by the Dreisam Valley (315–460 m a.s.l.), a flat valley filled with glacial deposits up to 50 m thick. Here, land use is a mix of settlements and agricultural land, with winter wheat and silage corn as the dominant crops. Arable land accounts for 2% of the total catchment area (CORINE Land Cover, 2018). In the lower



Dreisam Valley, the Ebnet waterworks are situated, abstracting 9 million m³ of groundwater per year for the water supply of the city of Freiburg. Further groundwater export in the aquifer below the Dreisam gauge was estimated to 60 mm (Didszun and Uhlenbrook, 2008). (ii) The mountainous Black Forest surrounding the Dreisam Valley is typified by the 40 km² Brugga sub-catchment. Here, 75% of the area is covered by steep, forested slopes with up to 62° inclinations. Grazing is limited to rounded mountaintops (20% of the area) and narrow valley floors (5%). The Brugga sub-catchment encompasses the 15.2 km² Talbach sub-catchment, which is largely designated as a nature conservation area.

## 2.2 Hydrological Processes in the DRC

TFA is a highly mobile anion whose transport is closely linked to hydrological pathways. We correlated TFA concentrations to environmental tracer data indicative of the active flow system. Three main streamflow components originating from different runoff generation processes were identified in the DRC by chemical (major ions, silica, and Hydrofluorocarbons - HFCs) and isotopic tracers (²H and ¹⁸O) (Uhlenbrook et al., 2002).

The first component is event water, which may account for up to 50% of streamflow during storm events. It is generated by Hortonian Overland Flow (HOF) from impervious surfaces, such as roads, rock outcrops, or urban areas, and by Saturation Overland Flow (SOF) from saturated areas, including wetlands and riparian zones.

The second component is shallow groundwater from the hillslopes, exhibiting an estimated mean transit time of approximately 2 to 3 years. Shallow groundwater accounts for the major fraction (approximately 70%) of runoff in the Brugga sub-catchment. It is exemplified by the Zipfeldobel spring, which primarily consists of this component and was as an additional sampling point.

The third component is deep groundwater originating from the crystalline bedrock, with an estimated mean transit time of approximately 6 to 9 years. This component contributes to approximately 20% of overall streamflow, but its proportion increases during prolonged drought periods (Uhlenbrook et al., 2002).

Subsequent investigations extended this general concept of runoff generation by observing Subsurface Storm Flow (SSF) on steep hillslopes. During discharge peaks, SSF contributed up to 50% of the streamflow in a catchment in the vicinity of the DRC (Bachmair and Weiler, 2014). Then, a network of pores interconnects to a system of hydrological flow paths that respond to precipitation according to its connectivity. Thus, SSF intensity depends both on soil moisture and event magnitude. Although event water may travel rapidly through SSF, a considerable portion of the mobilized water consists of "old" water already present in the flow system (Kienzler and Naef, 2008). Additionally, Wenninger et al. (2004) observed the effects of piston flow. They found that rapidly moving water from hillslopes exerted pressure on a shallow, confined valley aquifer, pushing old water into the Talbach. However, such conditions only prevail in valleys that were glaciated during the last ice age. These areas comprise only a small part of the upper Brugga catchment, and we do not consider this process to play a significant role in streamflow generation within the entire DRC.



### 2.3 Sources of TFA in DRC

Atmospheric TFA is a transformation product of volatile precursors such as HFC-134a and HFO-1234yf, and reaches the Earth's surface through wet deposition (Freeling et al., 2020; Franklin, 1993). Concentrations are typically enriched in samples from low precipitation volumes, necessitating precipitation volume weighting for representative input concentrations. Furthermore, atmospheric TFA deposition is higher during the summer (Freeling et al., 2020); therefore, measurements of at least one year are required to account for seasonality. In the DRC, we measured precipitation input at the St. Wilhelm climate
station for two years.

In the DRC, most arable land is located in the Dreisam valley near the catchment outlet (Figure 1). There, in addition to the ubiquitous atmospheric input, TFA can additionally be emitted from terrestrial sources:

    i)    Fluorinated compounds containing carbon-bound –$CF_3$ groups potentially degrade to TFA (Scheurer et al.,
2017; Sun et al., 2020). The surge in the use of such pesticides (Ogawa et al., 2020; Alexandrino et al., 2022)
125         could result in high TFA releases from agricultural areas.

    ii)    Agricultural land receives additional TFA inputs from organic fertilizers. Concentrations of TFA in liquid
manure and biogas digestate range between tens and hundreds of $\mu g\ L^{-1}$ (n = 3) (German Environment Agency,
2023). This is due to the accumulation of TFA in plants and foodstuffs (Ghisi et al., 2019; Scheurer and Nödler,
2021). Spreading of organic fertilizers is prohibited in Europe during winter (Liu et al., 2018) and pesticides are
130         mainly applied during the growing season. Hence, the input pattern of pesticides and liquid manure presumably
follows the annual pattern of atmospheric TFA deposition. Since data on the exact amounts of applied
pesticides and manure are missing, we quantified the relevance of diffuse TFA input from agriculture through a
mass balance using weekly TFA concentrations in the Dreisam River and samples from the aquifer beneath.

Scheurer et al. (2017) measured elevated TFA concentrations in the effluents of wastewater treatment plants (WWTPs). Three
small WWTPs with a capacity of 14650 population equivalents are located in the DRC, discharging on average about 17 L s$^{-1}$
(see Fig. 1). We sampled all three WWTP effluents to quantify the TFA introduced into the Dreisam river through wastewater.
Apart from precipitation, agriculture, ski wax, and wastewater, the direct release of TFA from fluorochemical industries is
reported in the literature (Scheurer et al., 2017). Since no such sites are located in the Dreisam catchment, this source was not
considered in this study.

### 2.4 Sample Collection

We collected weekly streamflow samples from the riverbank at a fixed position with turbulent flow. On the same day, we
retrieved precipitation samples: rainwater was stored in a polyethylene (PE) tank located below the funnel, which was emptied
after sampling. Furthermore, we obtained spring samples from the hillslope spring as a weekly grab sample. All samples
(n = 479) were filled in 100 ml brown glass bottles for major ion analysis and stored at 4°C in the dark for up to three weeks.



The same number of TFA samples were collected in parallel in 50 ml polypropylene centrifuge tubes (Greiner, Kremsmünster, Austria) for subsequent analysis. Storage time was up to four months, but a reanalysis of samples over one year (n = 3) confirmed that TFA concentrations remained stable during storage. The Ebnet waterworks collected groundwater samples from eight deep wells penetrating the Dreisam Valley aquifer. Drinking water in the eastern part of the city of Freiburg is

supplied by the same wells. Therefore, we compared the mean TFA value from the wells to drinking water values. The comparison suggested that groundwater-TFA levels remained constant over the two-year study period (compare Table 4). Three samples of the effluents were taken from the WWTPs in St. Peter, Hinterzarten, and St. Märgen to determine the WWTPs' input. A detailed description of all sampling sites is provided in Table A1. We measured conductivity ($C$), water temperature ($T$), and pH using handheld meters (LF325, pH330, WTW, Weilheim, Germany) during sampling in the rivers

Dreisam and Brugga, as well as in the Zipfeldobel spring. At the River Talbach, no measurements were taken due to the site's inaccessibility.

## 2.5 High-Pressure Liquid Chromatography-Tandem Mass Spectrometry (HPLC MS/MS) TFA Analysis

Samples were transferred into 1.5 ml Eppendorf tubes and centrifuged for 15 minutes at 18,000 $g$ (Centrifuge 5417, Eppendorf,

Germany). A 1 ml aliquot was transferred into LC vials (Isera Düren) and moved to the pre-cooled (4°C) Autosampler. Ion exchange liquid chromatography-electrospray tandem mass spectrometry (LC-ESI MS/MS) analysis using a Shimadzu LC-AD20 coupled with API 5500 Q Trap triple-quadrupole mass spectrometer (Applied Biosystems/MDS Sciex Instruments, Concord, ON, Canada) with an electrospray interface operating in negative ionization mode was applied for the detection of TFA (Table xy for MS/MS conditions) The injection volume was set to 50 µl, with separation performed on an IonPac AS17-

C column (2 × 250 mm) and an IonPac AG17-C guard column (2 × 50 mm) (both from Thermo Fisher Scientific, Waltham, USA). The column compartment was kept at 40 °C and a flow rate of 0.28 ml/min.

The analysis was conducted using an ultra-pure water solution containing 50 mM ammonium hydrogen carbonate (Honeywell Fluka™) (A) and methanol (ROTISOLV® ≥99,98 %, Ultra LC-MS Grade Carl Roth) (B) as eluents. The used columns were pre-equilibrated with the starting conditions of 20% A, and all samples were analyzed using the following gradient: 20% of

eluent A (0- 1 min), 20-50% A (1-10 min), 50-20% A (11- 16 min). Calibration was performed using a precisely prepared dilution series of TFA standards (pure trifluoracetic acid, obtained from Sigma-Aldrich) in ultra-pure water. UltraPure $H_2O$ (18.2 MΩ cm) was prepared in-house with a Milli-Q® Direct Water Purification System (Thermo-scientific®). The method is based on the method used by Scheurer et al. (2017).

Samples were measured in duplicate from the same vial and averaged, with a mean relative standard deviation of 1.9% over

all duplicates, calculated as described in (Synek, 2008). Each sampling batch (n = 12) included two MilliQ® blanks, prepared with the same procedure as the other sample in the batch to ensure the purity of the chemicals used, as well as at least three in-house standard samples (0.2, 0.29, 0.45, 0.5, 1.5, 3, and 6 µg/L) to ensure reproducibility (mean absolute percent error, MAPE,



8%). A limit of quantification (*LOQ*) of 0.08 µgL$^{-1}$ was determined according to DIN 32645 from the standard deviation (*SD*) of blank values.


**Table 1 MS Parameters**

| Compound | Precursor ion | Product ion | Declustering potential | Entrance potential | Collision energy | Cell exit potential |
|---|---|---|---|---|---|---|
| | [m/z] | | [V] | [V] | [V] | [V] |
| TFA | 112,9 | 68,9 | -8 | -12 | -18 | -12 |
| TFA-$^{13}$C$_2$ | 114,9 | 69,9 | -8 | -12 | -18 | -12 |

Because contamination during sample preparation is a concern, all used laboratory equipment (for example, Falcon tubes, Tips, and Eppendorf tubes) was washed with UltraPure H$_2$O and/or Ultrapure MeOH and analyzed by the method described
above. None of the equipment used in the laboratory showed relevant background levels of TFA. Consequently, blank correction of the analytical results was not required.

**2.6 Ion Chromatography (IC) Major Ions and Cavity Ring down Spectrometry (CRDS) Isotope Analysis**

From the 100 ml glass bottles, 5 ml aliquots were filtered using 0.45 µm syringe Millipore filters (VWR, Darmstadt, Germany)
and analyzed for major anions (chloride, nitrate, sulfate) and cations (sodium, potassium, magnesium, calcium) via ion chromatography (DIONEX ICS-1100, Thermo Fisher Scientific, Waltham, USA). For separation, we used an IonPac AS17-C column (4 × 250 mm) and an IonPac AG17-C guard column (4 × 50 mm) (both from Thermo Fisher Scientific, Waltham, USA). The supplier reported a precision of 5% and a quantification (LOQ) limit of 1 mg/L. Additionally, 1 ml aliquots were analyzed for stable isotopes using Cavity Ring-Down Spectroscopy with a L2130i System (Picarro, Santa Clara, USA),
achieving supplier-defined precisions of 0.16‰ for $^{18}$O and 0.6‰ for deuterium.

**2.7 Analysis of the TFA Time Series**

We obtained 15-minute discharge data from the Regional Council Freiburg for the Brugga River and the State Institute for Environment Baden-Württemberg (LUBW) for the Dreisam. At Zipfeldobel, a flow meter (IFC 010 System, KROHNE, Duisburg, Germany) was installed to log average discharge every 10 minutes. Data gaps were filled by linear interpolation
between weekly manual discharge measurements (see Fig. B2). Weekly precipitation amounts were collected at the St. Wilhelm Climate Station. Hourly precipitation volumes were logged by a tipping bucket setup at the same station and aggregated to daily values. To ensure accurate data from the tipping bucket setup, we calibrated the data using the manual samples. Data gaps were filled with data from a nearby station without prior interpolation (DWD-Buchenbach, distance ~5





km). First, we visually inspected our time series and compared discharge, precipitation, and TFA concentrations during runoff
events. Two time periods were selected to highlight the influence of dry and wet conditions: 15 September 2023 to 30
November 2023 and 9 August 2024 to 20 October 2024. We correlated TFA concentrations with water temperature and
discharge, major ions, pH, and isotopic tracer concentrations to gain insights into the active flow system and the source areas
of TFA. We utilized the *correlation_matrix* function from the R-package Hmisc (Harrell, 2024) to calculate the Pearson
correlation coefficients and determine the corresponding significance levels (*p*-values).

## 2.8 Water and TFA Mass Balance

We calculated annual water balances for all catchments for the hydrological years 2023 and 2024, from November 1 to October
31. Thereby, we disregarded storage changes and assumed that all precipitation (*P, mm*) exited the catchment through stream
discharge ($Q_S$, mm), groundwater outflow ($Q_{GW}$, mm), or evapotranspiration (*ET*, mm). None of the analyzed time series
exhibited a trend (Mann-Kendall test on a daily aggregation, $p < 0.05 \wedge \tau \notin [-0.1, 0.1]$). The water balance can thus be expressed
as:

$$P - ET = Q_S + Q_{GW} \tag{1}$$

Mean annual precipitation volumes were extracted from the HYRAS precipitation interpolation product for all catchment areas
with a one square kilometer special resolution (Rauthe et al., 2013). Before interpolation, the data were corrected for wind and
wetting errors at the precipitation sampling stations. The Talbach, Brugga, and Dreisam catchment areas were obtained from
the LUBW. At the Zipfeldobel spring, we calculated the mean of all nine cells surrounding the pixel where the spring is located
to ensure a robust estimation. The water balance method estimated the catchment area of the spring ($A_{ZI}$, mm$^2$) from annual
discharge ($Q_{Zi}$, mm$^3$) and ($ET_{Zi}$, mm):

$$A_{ZI} = \frac{Q_{ZI}}{P_{ZI} - ET_{ZI}} \tag{2}$$

*ET* was estimated from the 600 mm Brugga catchment average (Uhlenbrook et al., 2002) and reduced according to (Liu et al.,
2012) for a north-facing slope of 30° to 300 ± 100 mm. Annual discharge volumes were calculated as the mean from measured
discharge data. Because the Dreisam discharge gauge is located above the Dreisam aquifer, water exits the catchment as
groundwater below the gauge. Groundwater flow ($Q_{GW}$, m$^3$) was estimated using Darcy's Law, from the hydraulic conductivity
($k_f$, m s$^{-1}$), the hydraulic potential gradient ($\Delta h/\Delta L$, dimensionless), and the water-filled cross-sectional area of the aquifer (A,
m$^2$):

$$Q_{GW} = k_f \times A \times \frac{\Delta h}{\Delta L} \tag{3}$$

Appendix C provides details on groundwater flow calculation. In the Brugga and Talbach catchments, groundwater flow could
be excluded due to a lack of significant aquifers beneath the gauges.

By multiplying water balance volumes ($V_i$, m$^3$) (Eq. 1) with volume-weighted TFA concentrations ($c_i$, g L$^{-1}$), we established
the mass ($m_i$, kg) balance, assuming no transfer of ionic substances to the vapor phase:

$$c_P \times P = c_{Q_S} \times Q_S + c_{Q_{GW}} \times Q_{GW} \tag{4,}$$





$$m_i = c_i \times V_i \tag{5},$$

$$m_P = m_S + m_{GW} \tag{6},$$

where $m_P$ is the TFA input through precipitation, $m_S$ is the TFA output through streamflow or spring discharge, and $m_{GW}$ is the export through groundwater (all in kg), which is only relevant for the Dreisam main catchment. The individual components of

the mass balance were calculated for the hydrological years 2023 and 2024, with the corresponding weighted concentrations and volumes. Furthermore, an additional mass balance was established using a 30-year mean precipitation input, a 11 to 30-year mean discharge and groundwater output, and the two weighted mean TFA concentrations. This balance is referred to as the average annual balance. The time spans on which the average year calculations are based, are listed in Table A1. This approach quantifies retained TFA in the sub-catchments and the diffuse agricultural input in the main catchment. Due to a lack

of recent discharge data, the balance for the Talbach was only possible for the average year. The historical discharge data showed a high correlation ($R^2 = 0.92$) to the Brugga discharge. We therefore used Brugga discharges as weights to calculate the Talbach mean weighted TFA concentration. The groundwater TFA concentration was calculated as the mean of eight deep-well TFA measurements. Precipitation TFA input was calculated as a weighted mean using the weekly precipitation data from the St. Wilhelm climate station.

**2.9 Agricultural TFA excess**

For comparison with existing literature, we calculated the agricultural surplus per unit of arable land by dividing the total amounts of agricultural TFA input by the area of agricultural land. We used land cover data from the CORINE Land Cover (CLC) project (reference year: 2018) as described in Joerss et al. (2024). In the Dreisam main catchment, the excess per area ($m_{ex}$, kg km$^{-2}$) was calculated from average yearly values of the groundwater and streamflow outputs ($m_{GW}$ and $m_{QS}$, kg),

precipitation input ($m_P$, kg) and arable land area ($A_{arable}$, km$^2$):

$$m_{ex} = \frac{m_{GW} + m_{QS} - m_P}{A_{arable}} \tag{7}$$

**2.10 Calculating ET based on TFA-concentrations**

Assuming that TFA is not permanently retained, we calculated $ET$, based on weighted TFA concentrations in discharge ($c_{dis}$, µg L$^{-1}$) and precipitation ($c_p$, µg L$^{-1}$) and the mean groundwater concentration ($c_{GW}$, µg L$^{-1}$) along with the annual streamflow

($Q_S$, mm) and groundwater ($Q_{GW}$, mm):

$$ET = Q_S \times \left(\frac{c_{dis}}{c_P} - 1\right) + Q_{GW} \times \frac{c_{GW}}{c_P} \tag{8}$$

We compared the TFA-based $ET$ to the values calculated from the water balance to test the feasibility of this method.



**2.11 Error Estimation**

We propagated the error of the individual measurements through the calculations described in the chapter above. Appendix H describes the error propagation calculations in detail.

**3. Results**

**3.1 Visual analysis of two years' time series**

From a two-year TFA time series of the St. Wilhelm climate station, Zipfeldobel Spring, the Brugga, Talbach, and Dreisam Rivers, we observed the following (Fig. 2):

1. TFA levels in precipitation exhibited a seasonal pattern, with high values in summer and lower levels in winter.
2. TFA concentrations in the Dreisam River were generally higher than those observed in the sub-catchments and the
275         Zipfeldobel spring. Despite low precipitation levels, the highest TFA levels in the Dreisam River were observed during the 2023/2024 winter.
3. TFA concentrations increased with discharge, particularly noticeable in the Dreisam River. This effect is also evident in the Brugga and Talbach Rivers, but less distinct at the Zipfeldobel Spring.
4. At the end of drought periods in late summer, TFA concentrations in the Dreisam reached levels comparable to
280         those in the Zipfeldobel Spring and the Brugga and Talbach Rivers.



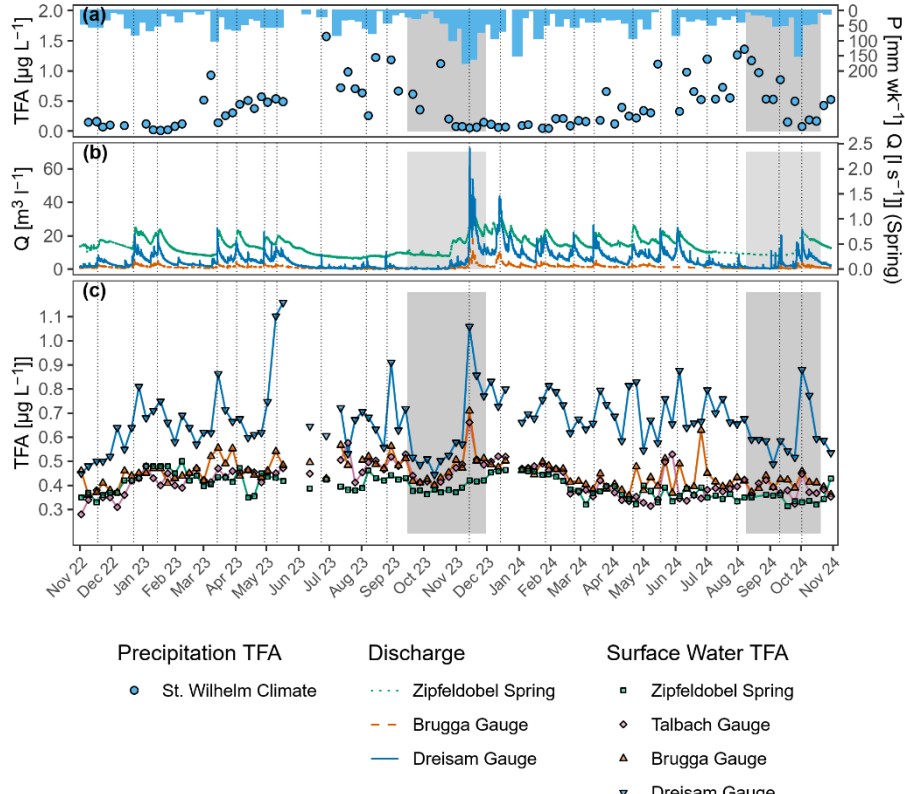

**Figure 2: Time series of precipitation, discharge, and TFA concentrations. Grey areas are highlighted in Figure 3. (a) Weekly precipitation is plotted against the corresponding weekly TFA concentration in rainwater at the St. Wilhelm station. (b) Discharge for Dreisam, Brugga, and Zipfeldobel was plotted. For Talbach, no current discharge data were available. (c) Weekly TFA concentration measurements in four surface waters of the DRC. Vertical dotted lines indicate discharge events.**






## 3.2 Events with changing flow conditions

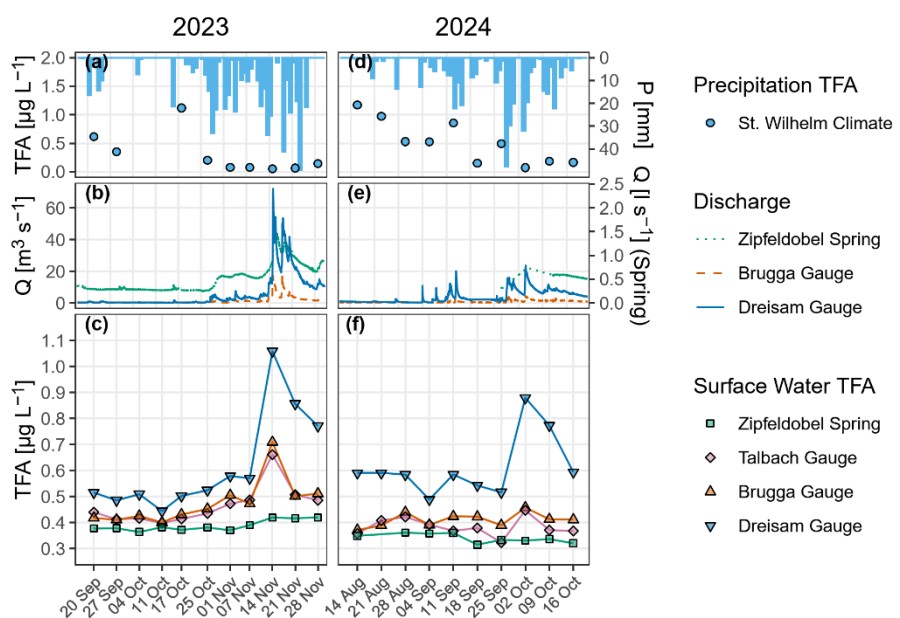

**Figure 3: Zoom into late summer/early autumn 2023 and 2024 periods. Due to a data gap, the Zipfeldobel spring discharge is not shown before 25 September 2024 (e).**

We selected two time periods in late summer to understand changes in TFA streamflow levels under varying flow conditions (see Fig. 3). Both periods transitioned from low-flow to high-flow conditions. In 2023, increasing precipitation depleted of TFA (a), resulted in increased discharge (b) and TFA levels (c) in the Dreisam, Brugga, and Talbach rivers. The springs' discharge reaction was delayed, and TFA levels remained almost constant. Contrasting this behavior, the main TFA peak was reached when discharge increased drastically on November 14th in all streams. The highest TFA concentration was observed

for the Dreisam River. Talbach and Brugga showed less pronounced TFA spiking. In 2024, minor precipitation events (d) resulted in a slight increase in discharge (e), which led to a corresponding rise in TFA levels on September 11. Until the next major precipitation event on the 25th of September, TFA levels decreased concurrently with discharge. The event then led to a doubling of TFA concentrations in the Dreisam River but only to minor peaks in Talbach and Brugga. The TFA level at the Zipfeldobel spring remained constant, despite the precipitation water exhibiting TFA concentrations below the Zipfeldobel

levels.

## 3.3 Correlation with water parameters

In addition to TFA concentrations, we recorded time series for $T$, p$H$, $C$, major ions, and stable water isotopes (see Fig. F1-5), and correlated them to the TFA concentrations in streamflow and precipitation (Table 2). In rainfall, TFA exhibited positive





correlations with all tracers, showing the strongest associations with potassium and stable water isotopes. We found positive
correlations in surface waters with nitrate, stable water isotopes, and discharge, which intensified from spring water to the
catchment outlet. The same was true for the negative correlation with deuterium excess. We also found weak to moderate
negative correlations with temperature, *pH*, and potassium concentration in Zipfeldobel Spring water and chloride in the
Dreisam River.

**Table 2: Pearson correlation coefficients of water parameters with TFA concentrations. Stars indicate the significance levels with * p ≤ 0.05, ** p ≤ 0.01 and ***p ≤ 0.001.**

|  | Dreisam Main C. | Brugga Sub C. | Talbach Sub-Sub C. | Zipfeldobel Spring | St. Wilhelm Precipitation |
|---|---|---|---|---|---|
| $Q$ or $P$ | 0.57*** | 0.42*** |  | 0.11 | -0.38*** |
| $T$ | -0.19 | -0.03 |  | -0.46*** |  |
| $C$ | 0.02 | -0.01 |  | -0.26* |  |
| $pH$ | -0.08 | -0.11 |  | -0.39*** |  |
| $NO_3^-$ | 0.59*** | 0.25* | 0.28** | 0.14 | 0.09 |
| $Cl^-$ | -0.27** | 0.11 | 0.10 | 0.07 | 0.37*** |
| $SO_4^{2-}$ | -0.04 | -0.24* | -0.18 | -0.17 | 0.30** |
| $Na^+$ | -0.16 | 0.17 | -0.00 | -0.18 | 0.33** |
| $K^+$ | 0.10 | 0.07 | 0.01 | -0.34*** | 0.66*** |
| $Mg^{2+}$ | -0.25* | -0.12 | -0.12 | -0.23* | 0.39*** |
| $Ca^{2+}$ | -0.14 | -0.11 | -0.11 | -0.18 | 0.33** |
| $\delta^{18}O$ | 0.58*** | 0.40*** | 0.39*** | 0.24* | 0.65*** |
| $\delta D$ | 0.53*** | 0.41*** | 0.44*** | 0.25* | 0.62*** |

## 3.4 Water balance and mean TFA concentrations

We calculated mean volume-weighted TFA concentrations for two consecutive hydrological years. During the observation
period, various climatic and hydrological conditions prevailed (see Table G1). The dry year in 2023 was followed by a wet
year in 2024 (compare Table 3). This resulted in a decrease of TFA concentrations in precipitation from 2023 to 2024, which
was offset by a slight mean increase of 5.3% in all three rivers (Table 4). Only concentrations in Zipfeldobel decreased by
7.0%. Volume-weighted mean annual TFA concentrations were highest in the Dreisam aquifer, followed by the Dreisam,
Brugga, and Talbach rivers, and lowest at the hillslope spring Zipfeldobel.

**Table 3: Water Balance - Precipitation Input, Groundwater and Surface Water Output. Values were calculated for the hydrological years 2023 and 2024 and the 30-year average (11 years at Zipfeldobel). Errors are given as standard errors.**





|  | Dreisam | | Brugga | Talbach | Zipfeldobel | St. Wilhelm |
|---|---|---|---|---|---|---|
| Input [mm] | *HYRAS* | | *HYRAS* | *HYRAS* | *HYRAS* | *Station* |
| Average | 1479 ± 67 | | 1806 ± 46 | 1851 ± 54 | 1659 ± 41 | 1826 ± 88 |
| 2023 | 1276 ± 185 | | 1531 ± 222 | 1500 ± 217 | 1506 ± 218 | 1569 ± 1 |
| 2024 | 1853 ± 269 | | 2170 ± 315 | 2153 ± 312 | 2106 ± 305 | 2473 ± 1 |
| | | | | | | |
| Output [mm] | *stream* | *groundwater* | *stream* | *stream* | *spring* | |
| Average | 687 ± 27 | 228 ± 23 | 1169 ± 38 | 1383 ± 55 | 1008 ± 63 | |
| 2023 | 422 ± 42 | 220 ± 109 | 775 ± 15 | | 907 ± 18 | |
| 2024 | 939 ± 94 | 242 ± 122 | 1280 ± 26 | | 1506 ± 29 | |

**Table 4: Volume-weighted mean TFA values for the hydrological years 2023 and 2024 and for an average of both years.**

|  | Dreisam | | Brugga | Talbach | Zipfeldobel | St. Wilhelm |
|---|---|---|---|---|---|---|
| $c_{TFA}$ [µg L$^{-1}$] | *stream* | *groundwater* | *stream* | *stream* | *spring* | *precipitation* |
| Average | 0.76 ± 0.05 | | 0.47 ± 0.03 | 0.44 ± 0.03 | 0.40 ± 0.01 | 0.33 ± 0.04 |
| 2023 | 0.73 ± 0.06 | | 0.46 ± 0.03 | 0.43 ± 0.03 | 0.42 ± 0.02 | 0.45 ± 0.07 |
| 2024 | 0.78 ± 0.06 | | 0.48 ± 0.04 | 0.45 ± 0.04 | 0.39 ± 0.02 | 0.25 ± 0.04 |
| Jun 2024 | | 0.81 ± 0.03* | | | | |
| Nov 2024 | | 0.64 ± 0.01** | | | | |
| Apr 2025 | | 0.84 ± 0.02** | | | | |

\* average from eight deep wells, ** drinking water from Ebnet waterworks


## 3.5 TFA mass balance and agricultural TFA excess

Combining volumes and concentrations, we calculated the TFA mass balance (Fig. 4). Lower precipitation levels decreased TFA mass input in all catchments from year 2023 to year 2024 by an average of 29% for all three stations with annual data. Contrasting the decreased input and the stable weighted mean TFA concentrations in streamflow, TFA export increased in 330 2024. The increase was particularly pronounced in the Dreisam River, where exports nearly doubled due to increased TFA loads in streamflow.

In addition to the separate balances for 2023 and 2024, we calculated mass balances for the average year in the DRC. Both sub-catchments and the spring showed evenly balanced import and export. Slight differences remained within the estimated error margins. The Dreisam River exported 48 ± 21 % more TFA. About one quarter was groundwater export. Based on the 335 observation from the headwater catchments, we did not assume any retention. Consequently, the Dreisam TFA export excess





translates to a TFA input of 11.4 ± 3.9 kg/km² from arable land. The TFA input from the three WWTPs within the catchment was minor (0.21 kg a⁻¹; Table D1) and could be neglected.

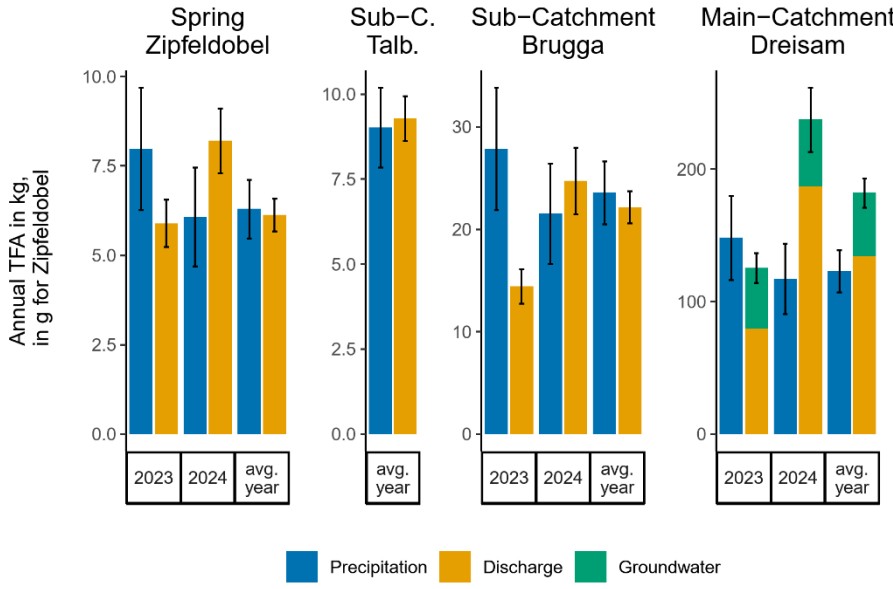

**Figure 4: The TFA mass balance for the spring Zipfeldobel, the nested sub-catchments Talbach and Brugga, and the main Dreisam catchment. In the Talbach catchment, no discharge data were available for either hydrological year, and therefore, only a balance for average conditions could be calculated. Whiskers indicate standard errors, calculated from bootstrapping standard errors for mean concentrations and volumes and consecutive Gaussian error propagation.**

**3.6 Calculation of ET from TFA concentrations**

Finally, we calculated the *ET* based on the deficit in the balance and from volume-weighted mean TFA concentrations in combination with the annual discharge (Table 5). *ET* from TFA at the Dreisam river was more than twice as high as the value obtained from the water balance. Brugga, Talbach, and Zipfeldobel showed comparable results within the estimated error margins.


**Table 5: ET values in mm, calculated from the water balance and TFA concentrations.**

| *ET* [mm] calculated from: | Dreisam | Brugga | Talbach | Zipfeldobel |
|---|---|---|---|---|
| Deficit in average annual water balance | 565 ± 50 | 644 ± 60 | 467 ± 72 | 300 ± 100 |
| TFA (Eq. 9) | 1411 ± 465 | 537 ± 241 | 521 ± 286 | 307 ± 214 |





## 4 Discussion

### 4.1 Quick discharge components control streamflow TFA concentration

In the Zipfeldobel hillslope spring, we hypothesize that the absence of TFA peaks in discharge stems from missing quick runoff components, as indicated by stable isotopes and water chemistry (compare Fig. F1-5), coupled with a slow discharge response to recent precipitation (Fig. 3). Previous studies also found that the portion of the direct discharge component was lower than 10% (Frey, 1999; Uhlenbrook, 1999). Consequently, TFA concentrations in precipitation did not noticeably affect spring TFA dynamics. Instead, weekly TFA concentrations remained nearly constant. Still, they showed a slight seasonal

pattern (Fig. 2). A weak positive correlation with stable water isotopes, enriched in summer precipitation, implied that the seasonality in spring discharge resulted from the seasonality of the input signal: Summer precipitation, enriched in TFA, was displaced from the flow system by more intense winter rainfalls. Therefore, any correlation with temperature may be coincidental, as the seasonal input signal shifts from input to discharge, and elevated TFA concentrations in spring discharge occur during winter. Potassium negatively correlated with TFA; however, concentrations were below LOQ and could not be

reliably interpreted. Furthermore, p$H$ showed a negative correlation with TFA. Prior findings indicated that TFA sorption to soils decreased with increasing p$H$ up to p$H$ 5 for soils with organic content smaller than 10% (Richey et al., 1997). The spring's p$H$ levels were above 6, and the primary flow path lies at the bedrock soil interface in the hillslope, where organic content is low. Consequently, sorption and desorption are likely not the driving processes explaining the correlation between TFA and pH at the Zipfeldobel spring.

Talbach and Brugga rivers mimicked the Zipfeldobel TFA seasonal pattern with a slight tendency towards higher levels. The main difference was that TFA concentrations in both streams responded to discharge peaks with elevated TFA concentrations. At some dates, stream TFA concentrations even exceeded those in precipitation, as indicated by the events shown in Fig. (3). Consequently, we hypothesize a temporal TFA storage that releases TFA into the quick runoff component, which is mainly SSF in the Brugga and Talbach catchments. A weak negative correlation with $Ca^{2+}$ and $Mg^{2+}$ implied the absence of

groundwater influence on TFA peaks in discharge. TFA concentrations peaked in conjunction with nitrate, which was also indicated by moderate positive correlations. Nitrate in forested areas, comparable to the Brugga catchment, mainly originates from the organic soil. Under dry conditions, the hillslopes are hydrologically disconnected, but when rewetting occurs, connectivity is restored. Then, SSF can transport nitrate from the soil to the stream (Lange and Haensler, 2012; Bachmair and Weiler, 2014). The parallel dynamics of TFA and nitrate suggested a common source and a similar transport mechanism.

Therefore, our hypothesis of a temporal TFA storage, which is most likely associated with organic soil, seems valid. Significant effects of TFA concentrations exceeding one µg g$^{-1}$ in soils on litter decomposition (Xu et al., 2022) warrant additional investigation. Next to nitrate, TFA concentrations correlated with stable water isotopes in Talbach and Brugga river discharge. Stable isotopes are enriched in summer precipitation due to higher temperatures, as are TFA concentrations due to increased



UV radiation (Franklin, 1993; Freeling et al., 2020). Therefore, simultaneous peaks of stable water isotopes and TFA suggest that water with high TFA concentrations was introduced to the system during the summer.

We argue that elevated TFA concentrations in the main catchment were caused by the use of PPP on the arable land at the valley floor: The Dreisam exhibited higher absolute TFA concentrations than the Brugga and Talbach rivers and stronger correlations with Nitrate and stable water isotopes. In the main catchment, nitrate presumably originates from agricultural activities, and precipitation tends to be heavier in the lower parts of the catchment, where the arable land is located. 390 Consequently, water emanating from the agricultural Dreisam Valley should display a heavier isotopic signature and higher nitrate levels, ultimately adding to the correlations already observed at Talbach and Brugga. Therefore, we argue that elevated TFA concentrations and increased correlations with nitrate and stable water isotopes indicate agricultural TFA transported from the Dreisam Valley into the stream by quick runoff components. In reverse, during dry periods devoid of quick runoff components, TFA concentrations were comparable to Talbach and Brugga. Then, TFA levels dropped during prolonged 395 droughts suggesting a higher fraction of groundwater in streamflow during autumn in 2023 and 2024 (Fig. 3). An increased groundwater influence was also indicated by high $Mg^{2+}$ and $Ca^{2+}$ concentrations during both drought periods (Fig. F3) and in general by the weak negative correlation of both ions with TFA. Freeling and Björnsdotter (2023) found a similar pattern in an agricultural catchment in Saxony, Germany. They showed that high TFA concentrations in streamflow were in line with parallel peaks of flufenacet-ESA, a substance indicating the release of TFA from pesticide degradation. The catchment's 400 fraction of agricultural area was 72%, about 36 times as high as in the DRC. This might explain why TFA in Saxony showed TFA concentrations of 1-2 µg $L^{-1}$ during late summer and 7-10 µg $L^{-1}$ during winter, much higher concentrations than in the Dreisam river (late summer: 0.45 – 0.6 µg $L^{-1}$, wet conditions in winter up to 1.1 µg $L^{-1}$). Additionally, increased dilution in the DRC due to a higher precipitation amount and an increased discharge compared to the Saxony catchment might add to the difference.


## 4.2 TFA Mass balances reveal no retention in areas free of agriculture but a TFA excess in the agricultural main catchment

In the dry year 2023, we observed higher input than output in the TFA mass balance at all sampling locations. In the wet year 2024, this pattern was reversed. Then, precipitation volumes increased, but precipitation TFA concentrations dropped, 410 decreasing the overall precipitation input load. Freeling et al. (2020) found a correlation between TFA levels in precipitation and incoming solar radiation and attributed this to the atmospheric degradation of TFA precursors. Indeed, the incoming shortwave solar radiation measured at the St. Wilhelm station decreased by 23% from 2023 to 2024, explaining the average 29% decrease in TFA input mass. Consequently, our data suggest that the mass of TFA deposited in a catchment primarily depends on the radiation-driven degradation of atmospheric precursors and not on the amount of precipitation. Average 415 weighted concentrations decreased with increasing precipitation volume. Consequently, TFA mass input did not scale with the amount of rainfall. Therefore, multiplying average TFA concentrations with higher or lower precipitation amounts might lead



to an over- or underestimation of the TFA mass input. Consequently, we question the approach of scaling weighted mean TFA concentrations between different years. However, in the DRC case, the average year falls between the two measured years in terms of precipitation volume. Therefore, calculating the mass input for an average year as the product of the two-year weighted
mean and the average precipitation volume seems justified.

Contrary to the input, the TFA output increased in 2024. Most of the additional load stems from increased discharge, as weighted mean TFA concentrations in the rivers only rose slightly. Apparently, the effects of the previously discussed SSF TFA mobilization overruled the dilution effect of TFA-depleted event water in 2024. Regarding the average annual output masses, we observed two hydrological years with opposing wetness conditions, and mean concentrations only varied slightly.
In such a case, calculating the TFA output for an average year by multiplying average concentrations with the average discharge volumes appears reasonable. Therefore, we believe that this study's average year balance accurately reflects the average conditions.

Similar TFA in- and outputs during the average year suggested non-significant retention in the spring and in the two upper catchments. This observation disagrees with former studies, which claimed TFA retention in plants and soils (Likens et al.,
1997; Berger et al., 1997). Potentially, differences originate from the study design of both field experiments. Labeling with roughly a 1000-fold of today's annual background flux might have led to higher TFA uptake, and the sampling interval of below one year might not have captured effects like the release of previously taken up TFA from decomposing dead organic matter. With up to 1 mg kg$^{-1}$ dry weight (Freeling et al., 2022), leaves and needles might build up an enormous organic TFA pool. The decomposition of the organic matter might release TFA in the long term from soils, which is in line with the
observation from the previous chapter (correlation with nitrate and discharge because of SSF through the soil zone).

Contrasting the observation in the sub-catchments, we found excess TFA export for an average year in the main catchment, which contains arable land at the valley floor. We hypothesize that the excess TFA stems from the degradation of PPP containing TFA precursors (also compare Section 4.1). Roughly a quarter of the total TFA export was through groundwater, which in the Dreisam valley is recharged from rain and rivers like the Brugga. Comparing TFA concentrations, the groundwater
levels are above those of rainfall and rivers. Therefore, elevated groundwater levels support the assumption that the excess in TFA exports originates from agricultural activities.

Joerss et al. (2024) published a dataset estimating agricultural TFA input based on a European dataset regarding emissions from plant protection products. Normalizing the amount of agrarian TFA on the area of arable land yields values of 3.3 kg km$^{-2}$ for Freiburg and 4.5 ± 1.7 kg km$^{-2}$ for the entity of Baden-Württemberg, assuming a 100% molar yield of TFA from -CF$_3$
pesticides. The amount we found in the DRC, at 11.4 ± 3.9 kg km$^{-2}$, is more than twice as high. A potential explanation offers the application of manure, containing TFA, which has been imported with fodder into the DRC.

Another reason for overestimating the TFA excess lies in potential heterogeneities in the precipitation input. Patterns in TFA surface water concentrations (Cahill, 2022) and elevated concentrations in precipitation in the vicinity of cities (Freeling et al., 2020) were attributed to the distribution of TFA precursor molecules in the atmosphere. Whether those observations hold on
a meso-catchment scale remains unclear. The lower part of the DRC is located downwind of the city of Freiburg. Therefore,



elevated precipitation concentration in the Dreisam valley might be possible. Consequently, the spatial variation of input concentrations near Freiburg might explain some differences in the agricultural excess TFA amounts.

## 4.3 Evaluating the possibility of *ET* calculation from discharge and TFA concentrations

TFA (Eq. 9) calculated ET values aligned with values obtained from the water balance within the given error margins, except for the Dreisam river, where a surplus of TFA export led to an overestimated *ET*. Calculating *ET* from TFA concentration resulted in higher error margins than using the water balance. High fluctuations in TFA in precipitation were the primary source of error.

Overall, we expected an underestimation of *ET* calculated from TFA concentration, mainly because recent precipitation concentrations might overestimate the amount of TFA that entered the system in the past. The mean time shift between input
and outflow, also known as the mean residence time in the Brugga catchment, is in the range of 11 to 16 months (Uhlenbrook et al., 2002). The 3-4 fold increase of TFA concentrations in precipitation over the last 28-30 years (Freeling et al. 2020) translates to an annual growth between 4 and 5% based on the assumption of an exponential increase (German Environment Agency, 2021). Consequently, when comparing recent streamflow and precipitation concentrations to determine *ET*, we overestimate the input by 4-8%. Compared to the relative *ET* errors calculated from TFA, ranging from 45% at Brugga to 70%
at Zipfeldobel, this difference seems relatively small and might not show up in the TFA mass balance.

Rising TFA levels generally limit the number of comparable years needed to calculate a representative precipitation mean, and the interannual variability necessitates at least a few years for accurate prediction. Our results suggest that despite the high error margins, at least two years of observations produced reasonable ET estimates for all three sub-catchments in the DRC.

## 5. Conclusion

We identified the organic soil zone as a primary TFA storage, and SSF as the dominating process for transporting TFA to the river. However, a two-year mass balance provided no evidence of permanent TFA retention in the two mountainous sub-catchments and at a hillslope spring. Temporal accumulation was only observed during dry periods; thereafter, the export of retained TFA was enhanced during the following wet conditions. The absence of permanent retention allowed for the calculation of the agricultural surplus associated with the arable land in the main catchment. The TFA loads resulting from
farming activities were found to be significantly higher than values reported for the degradation of precursor PPP in literature (Joerss et al., 2024), underling the relevance of agriculture in the overall TFA pollution.

Our findings on limited TFA retention have a bearing on the use of TFA as a tool to estimate actual *ET*. This approach is applicable in catchments with solely atmospheric TFA input and mean residence times of 2-5 years. Main uncertainties arise from difficulties in finding representative precipitation TFA concentrations, as overall TFA concentrations are increasing. Our
results showed that *ET* could reliably be estimated from just two years of TFA concentrations in this particular case. Therefore,





*ET* calculation from TFA has the potential to substitute for the widely used chloride method (Wood and Sanford, 1995) in areas where chloride is introduced by road salt or geogenic chloride is present.

We generally advocate for a perspective on TFA as both a contaminant and a valuable tool for hydrological process research. Albers and Sültenfuss (2024) demonstrated the use of TFA concentrations in groundwater for age dating, and Lange et al.

(submitted) evaluated the potential of TFA as a new hydrological pollution tracer. In the future, elevated TFA concentrations in SSF, combined with higher-frequency sampling, could yield valuable insights into this runoff generation process, supposed that TFA is not retained in the long term, a fact which remains to be proven in other areas. In the DRC, the seasonal input signal and the rising concentrations in precipitation qualified TFA as a tool for transit time estimation. Accordingly, we call for similar investigations in other catchments.






**Appendix A**

**Table A1: Overview of all sampled locations, intervals, parameters, and timely resolution.**

| | Type | Sampling Interval | Parameters | Remark |
|---|---|---|---|---|
| St. Wilhelm | Precipitation | Weekly, 2022-11-01 – 2024-11-01 | $V$, $Cl^-$, $NO_3^-$, $SO_4^{2-}$, $TFA^-$ $K^+$, $Na^+$, $Ca^{2+}$, $Mg^{2+}$, | Additional measurements: wind speed and direction, $T$, incoming solar radiation |
| Zipfeldobel | Spring Discharge | Weekly 2022-11-01 – 2024-11-01 | $Q, T, C, pH$, $Cl^-$, $NO_3^-$, $SO_4^{2-}$, $TFA^-$ $K^+$, $Na^+$, $Ca^{2+}$, $Mg^{2+}$ | Discharge for 2004 and 2014-2024, 10 min |
| Talbach | River Discharge | Weekly 2022-11-01 – 2024-11-01 | $Cl^-$, $NO_3^-$, $SO_4^{2-}$, $TFA^-$ $K^+$, $Na^+$, $Ca^{2+}$, $Mg^{2+}$ | Discharge data from 1980 until 2009, daily |
| Brugga | River Discharge | Weekly 2022-11-01 – 2024-11-01 | $Q, T, C, pH$, $Cl^-$, $NO_3^-$, $SO_4^{2-}$, $TFA^-$ $K^+$, $Na^+$, $Ca^{2+}$, $Mg^{2+}$ | Discharge data from 1995 until 2024, 15 min |
| Dreisam | River Discharge | Weekly 2022-11-01 – 2024-11-01 | $Q, T, C, pH$, $Cl^-$, $NO_3^-$, $SO_4^{2-}$, $TFA^-$ $K^+$, $Na^+$, $Ca^{2+}$, $Mg^{2+}$ | Discharge data from 1995 until 2024, 15 min |
| St. Peter | WWTP Outflow | Point Sample 2024-03-27 | $TFA^-$ | |
| St. Märgen | WWTP Outflow | Point Sample 2024-03-27 | $TFA^-$ | |
| Hinterzarten | WWTP Outflow | Point Sample 2023-03-05 2023-03-30 | $TFA^-$ | |
| Ebnet | Groundwater | Point Sample of 8 wells June 2024 | $TFA^-$ | |





**Appendix B**

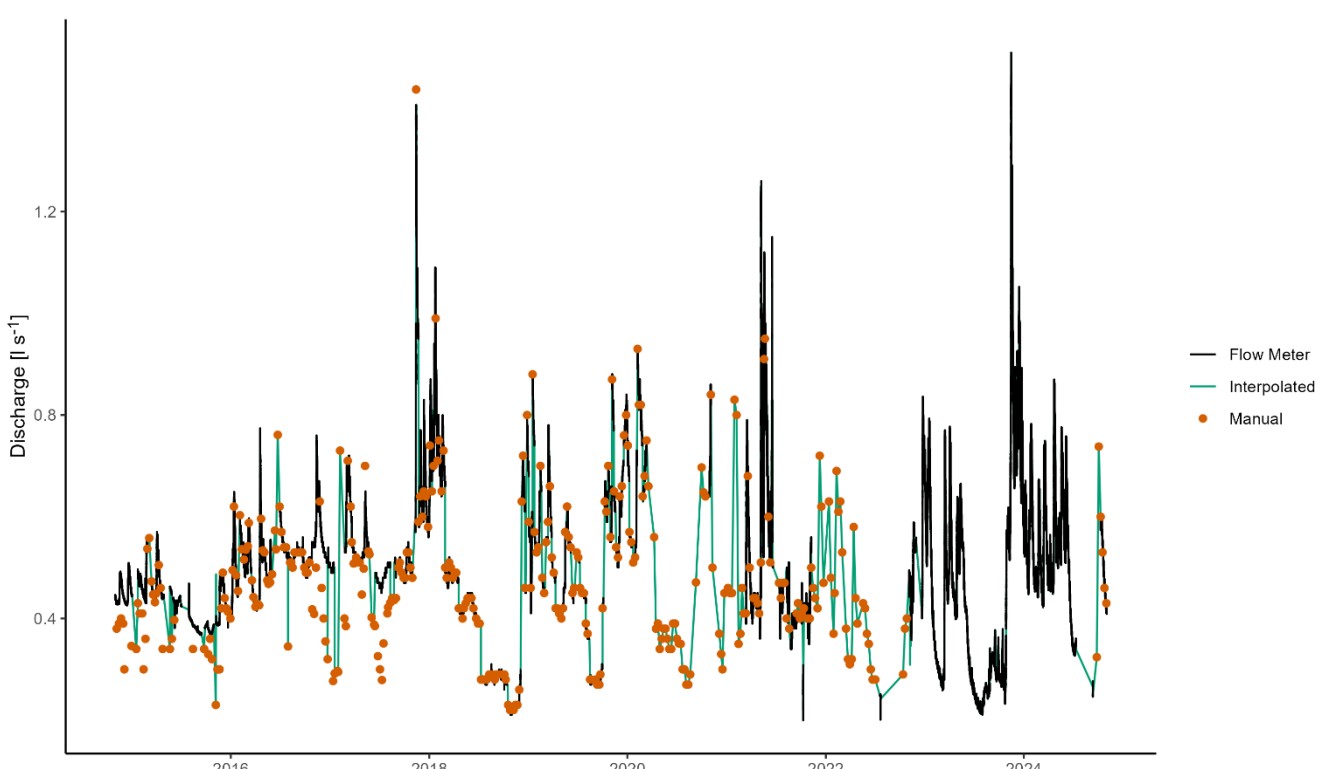

**Figure B1: Discharge time series measured by the inductive flow meter at Zipfeldobel (black), manual measurements used for linear interpolation (orange dots), and interpolated values (green).**


**Appendix C**

In the Dreisam catchment, groundwater flow was calculated using Darcy's law. $k_f$-values of three well-duplets and two individual wells (4.75e-4, 5.67e-4, 9.39e-4, 3.65e-3, 7.27e-3, 2.17e-3, 4.03e-3, 3.72e-2, distance < 1km from the gauge) were obtained from a local engineering office. The average $k_f$-value of $7.04 \times 10^{-3}$ m s$^{-1}$ was calculated. Hydraulic potential and water-filled area were estimated using groundwater gauging stations located upstream and downstream (see Fig. C1, L1, and L2), supplemented by geological data from the ISONG product by Landesamt für Geologie, Rohstoffe und Berbau (LGRB) (see Fig. C2): We calculated annual mean water table depths at L1 and L2. The difference in the water tables divided by the sum of each vertical distance to the transect is the slope of the groundwater table. The water table at the transect is then calculated by adding the slope times the perpendicular distance L1-transect to the L1 groundwater table. Then, the water-filled area is calculated from the geological profile and the water table. For the hydrological years 2023 and 2024, 183 mm and 205 mm values were calculated. Based on 25 years of groundwater table data, the mean annual groundwater flow is 191 mm. We added 37 ± 2 mm due to groundwater pumping activity at the drinking water facility Ebnet, resulting in 220, 242, and 228 mm.



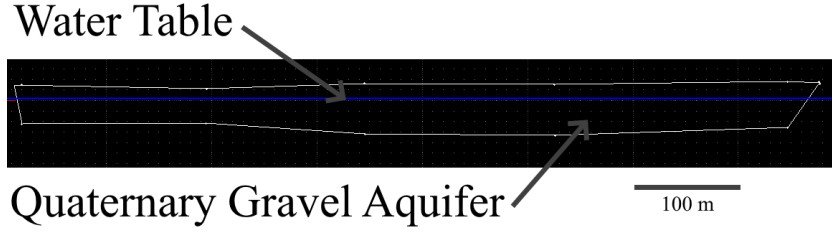


**Figure C2: Geologic profile of the aquifer below the Dreisam gauge.**

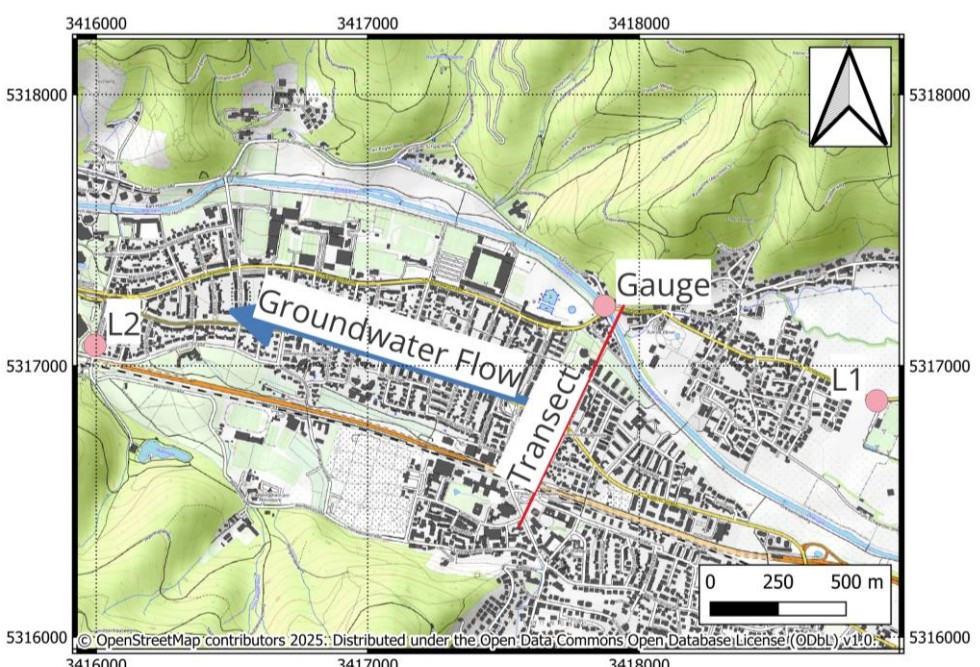

**Figure C1: DRC gauge area. Groundwater gauges L1 and L2, the transect on which a geologic profile was taken, and the main**
**groundwater flow direction (blue arrow).**





**Appendix D**

We compared historical data collected within the study catchment at five locations to grid point estimates from the HYRAS product (Fig. D1). The station data (Fig. D2) was not used in the interpolation algorithm and corrected as described in Richter 1995 before comparison (D1). The basic assumption was that the mean error from the stations represents the error of the whole interpolated area. At an annual aggregation level, we obtained a relative standard error of the estimate (rSEE) of (16%), a mean relative absolute error (MRAE) of 12.0%, and a bias of –51 mm. The bias was relatively small compared to all stations' mean HYRAS prediction of 1700 mm combined. Additionally, most of the bias stems from the Schauinsland (si) climate station, which is located on the windward slope of the mountain, leading to increased precipitation amounts. The model did not capture this effect. The error was normal distributed (Shapiro 1964, W = 0.98, p = 0.76). The distribution is shown in D3. Therefore, we used the rSEE as a standard error for later error calculations.

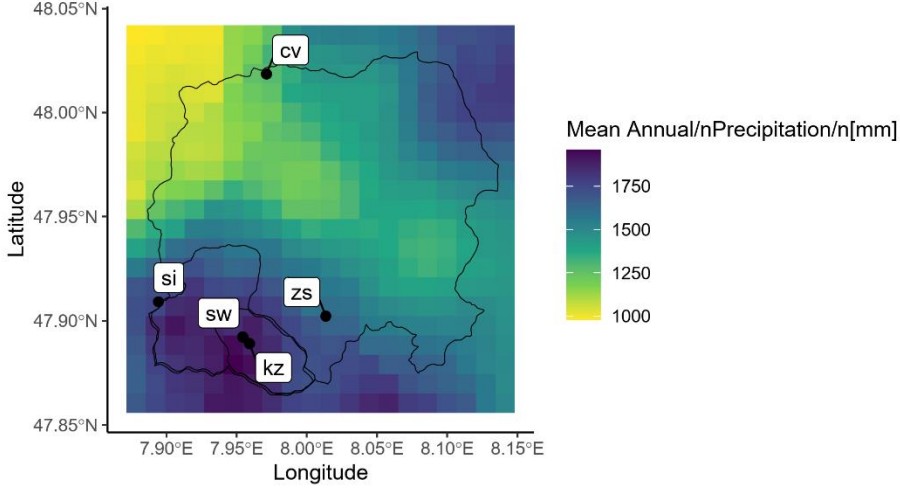

**Figure D1: shows the annual mean precipitation of the HYRAS interpolation product. Points indicate climate stations not included in the interpolation algorithm and used for error estimation.**





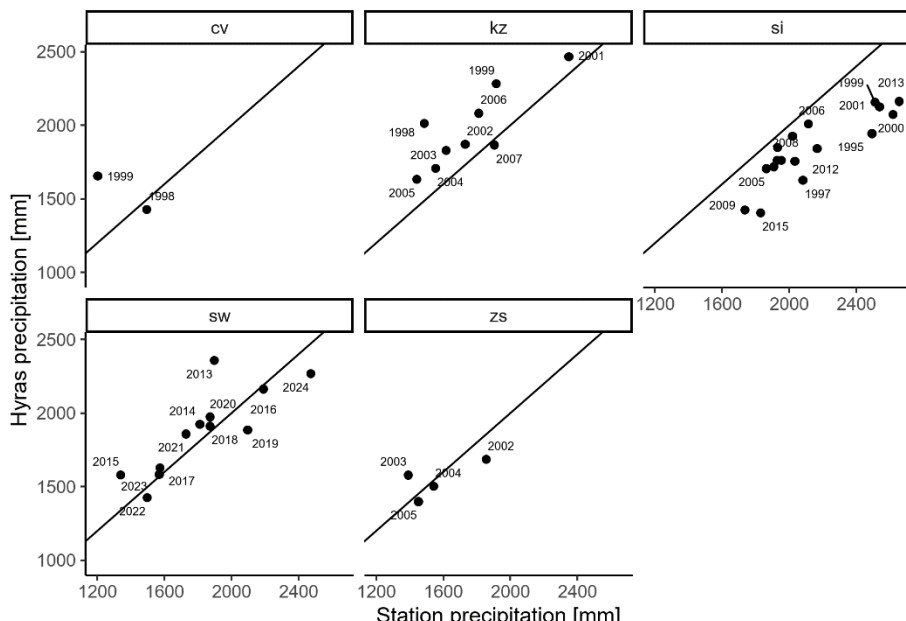


**Figure D2: compares annual HYRAS precipitation data with five climate stations' values. The black line indicates the perfect fit.**



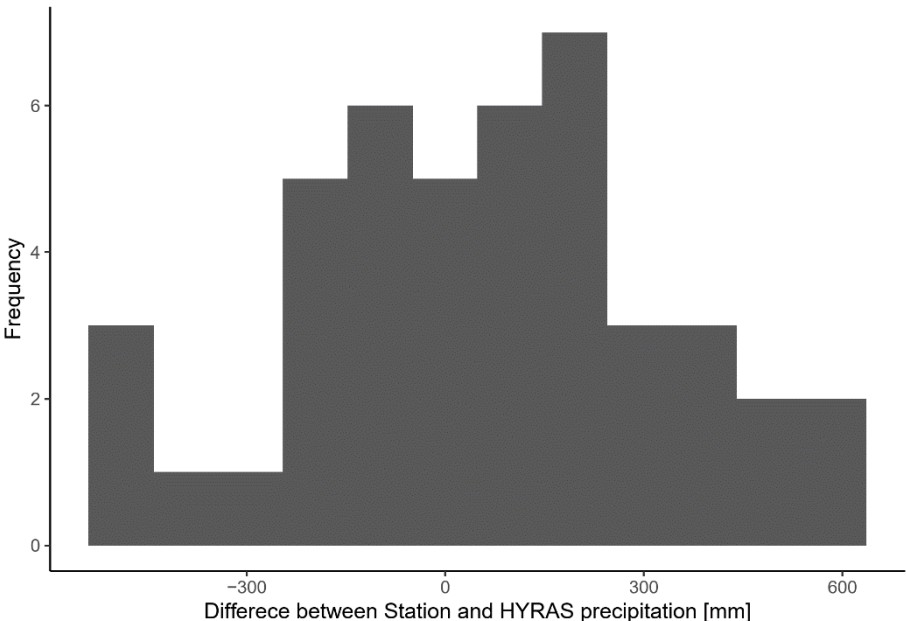

**Figure D3: shows the distribution of differences between the annual station and HYRAS precipitation data.**

**Table D1: Correction for wind and wetting errors and parameters necessary for correction for climate stations located in the Dreisam catchment.**

|  | Sampling Method | Average wind speed [m/s] | Exposition | Average Correction [%] |
|---|---|---|---|---|
| Conventwald (cv) | Tipping scale | 0.7 | Heavily sheltered | + 6.4 |
| Katzensteig (kz) | Tipping scale | 1.4 | Heavily sheltered | + 6.0 |
| Schauinsland (si) | Tipping scale | 3.0 | Moderately Sheltered | + 8.9 |
| St. Wilhelm (sw) | Manual | 1.1 | Heavily sheltered | + 3.1 |
| Zastler (zs) | Tipping scale | 0.9 | Heavily sheltered | + 5.9 |





**Appendix E**

**Table E1. Discharge, TFA concentrations, and loads at WWTPs in the Dreisam catchment.**

| Date | Location | Type | c(TFA) [µg L$^{-1}$] | discharge L s$^{-1}$ | annual load [kg] |
|---|---|---|---|---|---|
| 2024-03-27 | St. Maergern | Outflow | 1,19 | 3,19 | 0,12 |
| 2024-03-27 | St. Peter | Outflow | 1,07 | 5,43 | 0,18 |
| 2023-03-05 | Hinterzarten | Outflow | 0,49 | 8,68 | 0,13 |
| 2023-03-30 | Hinterzarten | Outflow | 0,5 | 8,68 | 0,14 |

We point-sampled all three wastewater treatment plants (WWTP) in the Dreisam catchment (E1). The total TFA load released from the WWTPs is 0.43 kg a$^{-1}$ year. The drinking water that ultimately goes to wastewater is mainly abstracted from springs in the area. Assuming a mean TFA concentration of 0.39 µg L$^{-1}$ (measured at the Zipfeldobel spring), 0.22 kg had been

introduced into the system before by precipitation and was already captured by the balance. Consequently, we subtracted that value and found a surplus of 0.21 kg a$^{-1}$ for all three WWTPs combined.



**Appendix F**



**Figure F1: Time series of isotopic tracer data recorded in surface waters for the hydrological years 2023 and 2024.**





**Figure F2: Time series of isotopic tracer data recorded in precipitation for the hydrological years 2023 and 2024.**



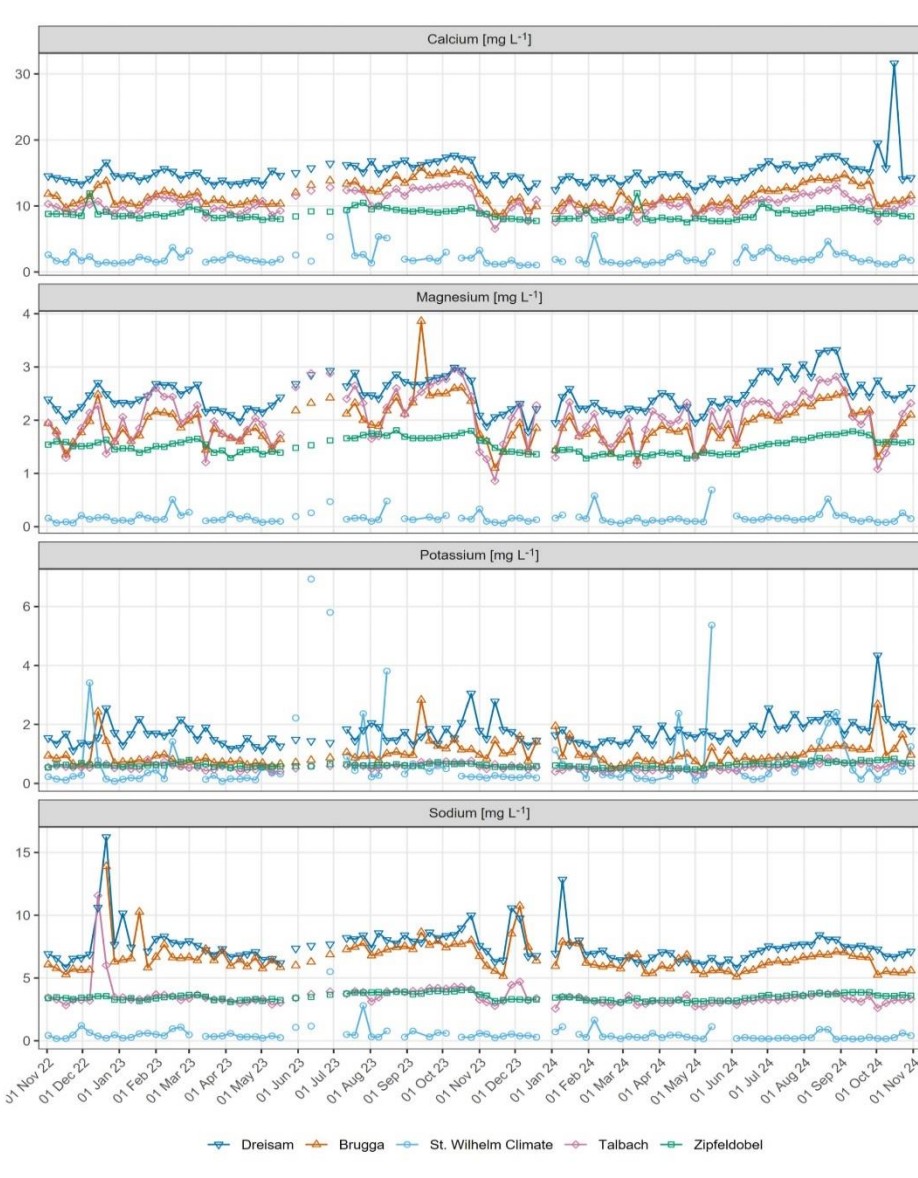

**Figure F3: Time series of cation tracer data recorded in surface waters and precipitation for the hydrological years 2023 and 2024.**





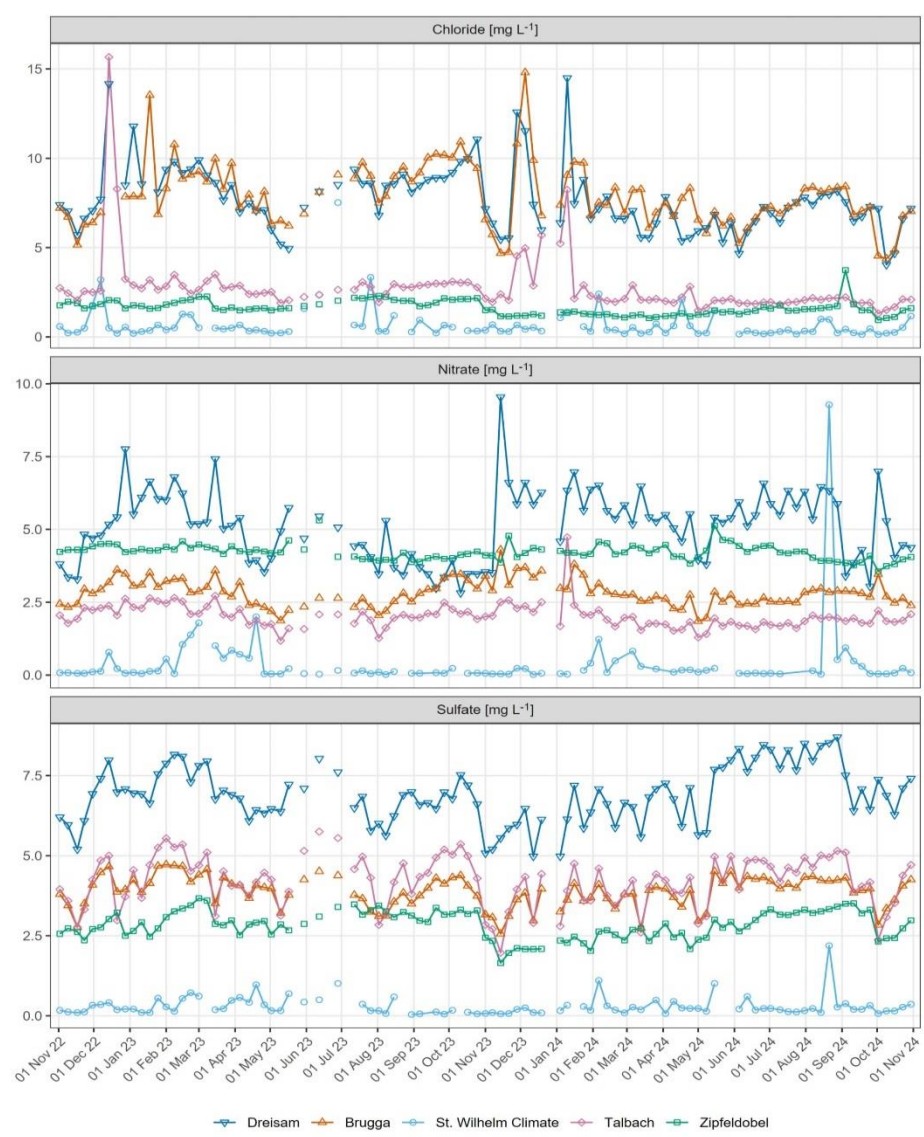


**Figure F4. Time series of anion tracer data recorded in surface waters and precipitation for the hydrological years 2023 and 2024.**

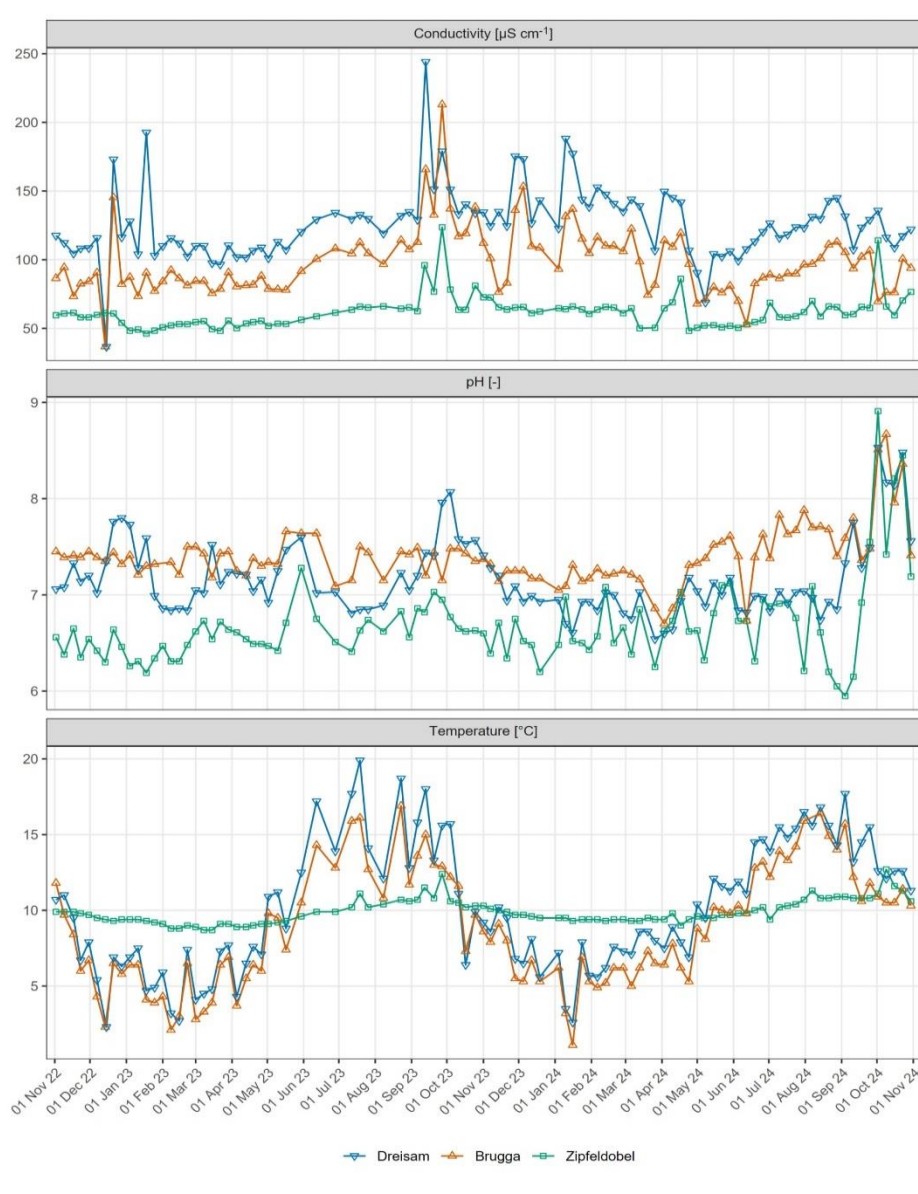

**Figure F5: Time series of physical parameters recorded in surface waters for the hydrological years 2023 and 2024.**




**Appendix G**

**Table G1: Exceedance probabilities for precipitation and discharge for the hydrological years 2023 and 2024 in the 30 years of recorded data (less for Zipfeldobel and Talbach).**

| Exceedance Probability [%] | | Dreisam | Brugga | Talbach | Zipfeldobel | St. Wilhelm |
|---|---|---|---|---|---|---|
| Input | | *HYRAS* | *HYRAS* | *HYRAS* | *HYRAS* | *Station* |
| 2023 | | 84% | 81% | 87% | 71% | 50% |
| 2024 | | 3% | 6% | 0% | 16% | 3% |
| | | | | | | |
| Output | *stream* | *groundwater* | *stream* | *stream* | *spring* | |
| 2023 | 88% | 73% | 96% | | 80% | |
| 2024 | 16% | 15% | 33% | | 10% | |




## Appendix H

## Error Estimation

### 1 Bootstrap error estimation

Errors for means and weighted means were bootstrapped. The bootstrap method utilizes random resampling with replacement to obtain more observations. We resampled the datasets 10,000 times, added random noise to the data according to the initial errors of the observations and weights, and computed the mean or the weighted mean for each resampling. From the yielded (weighted) means, we calculated the standard error.

### 2 Discharge

Discharge at the gauges Dreisam, Brugga, and Talbach is measured using a stage-discharge relationship. Therefore, errors occur in the stage measurement and the estimation of the rating curve. (Horner et al., 2018) estimated the median total uncertainty for eight French rivers at different time scales. Uncertainties ranged between 5% and 34% for hourly discharge measurement and between 1.4% and 10% for annual discharge, with most of the error stemming from the rating curve estimate. Based on those values, we assume 30% for the 15- and 10-minute discharge measurements and 10% for annual discharge. At

Zipfeldobel, the supplier reported a volume error of 0.5% for the magnetic inductive flow meter. A comparison revealed a CV of 0.49% between the manual discharge measurements and those obtained using the flow meter. Therefore, 0.5% was used as the error for individual discharge measurements. The high amount of missing data potentially influences the annual discharge volumes. As a result, the annual volume error could be significantly higher; we assumed 5 times the error of individual measurements, which results in a 2.5% error. At all locations, the uncertainty of the long-term mean discharge was calculated

from the annual discharges and their corresponding errors using the bootstrap method described in the previous chapter.




## 3 Precipitation

To estimate the error in the annual precipitation input predicted by the HYRAS model, we compared the HYRAS data with historical data from five climate stations located within the Dreisam catchment. The standard error of the annual estimate

($\sigma_{est} = 14.5\%$) (Eq. H1) was used as the standard error for subsequent error propagation. $d_i$ represents the distance between the estimate and the station value ($P_S$, mm), and $n$ is the sample size:

$$\sigma_{est} = \sqrt{\frac{\Sigma\left(\frac{d_i}{P_S}\right)^2}{n}}$$
(H1)

The reader is referred to Appendix D for a detailed description of the method. The long-term precipitation mean is calculated

by bootstrapping from annual precipitation values and their corresponding errors. Manual precipitation measurements at St. Wilhelm climate station were uncertain due to the manual volume measurement. Calculating the amount of precipitation results in an error ($\sigma_{P,weekly}$) of 0.1 mm for a weekly measurement. Later, we used this error to measure uncertainty for the weights when computing the volume-weighted mean TFA concentration in precipitation. The annual volume error ($\sigma_{P,annual}$, mm) is calculated as described in Eq. H2:

$$\sigma_{P_{annual}} = \sqrt{\sum \sigma_{P_{weekly_i}}^2}$$
(H2)

## 4 Groundwater Flow

Errors arise from the variance in the found $k_f$-values for the Dreisam aquifer (eight individual $k_f$ values from wells, all located within 1 km of the Dreisam gauge) and potential errors in the water-filled area. The error for a mean $k_f$-value $\sigma_{kf}$ was

bootstrapped. For the water-filled area (see Fig. C2), we assumed a random error $\sigma_A = 10\%$. Errors arising from water table depth measurements were estimated to $\sigma_{\Delta h} = 1$ cm. We propagated the Gaussian error for the single-year groundwater flow, neglecting errors in the distance measurements (QGIS measurement tool) due to their relatively small impact as described in Eq. (H3). We bootstrapped the error for the 25-year long-term mean.

$$\sigma_{Q_{GW}} = \sqrt{\left(\sigma_{k_f} \times A \times \frac{\Delta h}{\Delta L}\right)^2 \times \left(\sigma_A \times k_f \times \frac{\Delta h}{\Delta L}\right)^2 \times \left(\sigma_{\Delta h} \times k_f \times \frac{1}{\Delta L}\right)^2}$$
(H3)


## 5 Spring Zipfeldobel catchment area

The error of the mean Zipfeldobel spring catchment area ($\sigma_{A,ZI}$, m$^2$) is bootstrapped (see Sect. 2.10.1) from the calculated annual catchment areas. The catchment area's error for a single year is given by Eq. (H4):

$$\sigma_{A_{ZI}} = \sqrt{\left(\frac{1}{P_{ZI} - ET_{ZI}} \times \sigma_{Q_{ZI}}\right)^2 + \left(\frac{-Q_{ZI}}{(P_{ZI} - ET_{ZI})^2} \times \sigma_{P_{ZI}}\right)^2 + \left(\frac{-Q_{ZI}}{(P_{ZI} - ET_{ZI})^2} \times \sigma_{ET_{ZI}}\right)^2}$$
(H4)





## 6 TFA concentrations


The relative standard error of the analytical method ($\sigma_s$, %) is calculated from the relative distances ($rd_i$, %) of all the duplicated measurements from their mean by Eq. (H5) (Eckschlager, 1969):

$$\sigma_s = \sqrt{\frac{\sum rd_i}{2n}} \tag{H5}$$

Errors in the volume-weighted mean TFA concentrations in surface waters were bootstrapped using the standard error of the
TFA measurement (1.9 %) and the error of the 15-minute discharge values (30%). Errors for the weighted precipitation means were bootstrapped using the weekly precipitation (± 1 mm) and TFA values from the St. Wilhelm climate station. The mean TFA groundwater concentration error was estimated using bootstrapping of measurements from eight individual wells.

## 7 Errors of the mass balance's components

The propagation of the error for each component of the mass balance $\sigma_{mi}$ (precipitation input, streamflow output, and
groundwater output, kg) was calculated from the concentration $c_i$ and the corresponding Volume ($V_i$, m$^3$) and their corresponding errors ($\sigma_{ci}$ and $\sigma_{Vi}$, kg and g L$^{-1}$) according to Eq. (H6):

$$\sigma_{m_i} = \sqrt{\left(\sigma_{c_i} \times V_i\right)^2 + \left(\sigma_{V_i} \times c_i\right)^2} \tag{H6}$$

At Dreisam, the values for groundwater ($m_{GW}$, kg) and streamflow export ($m_S$, kg) were added together. The error for the sum of both export values ($\sigma_{m, ex}$, kg) was calculated according to Eq. (H7):

$$\sigma_{m_{ex}} = \sqrt{\left(\sigma_{m_{GW}} \times m_S\right)^2 + \left(\sigma_{m_S} \times m_{GW}\right)^2} \tag{H7}$$

## 8 Error of the agricultural TFA excess

The error of the agricultural TFA mass excess ($\sigma_{ex}$, kg) results from the error values of the groundwater and streamflow outputs ($m_{GW}$ and $m_{QS}$, kg), precipitation input ($m_P$, kg), and arable land area ($A_{arable}$, km$^2$) as shown in Eq. (H8). Because the CLC does not provide error metrics, an error of 10% was assumed for the arable land area.

$$\sigma_{m_{ex}} = \sqrt{\left(\frac{\sigma_{m_{GW}}}{A_{arable}}\right)^2 + \left(\frac{\sigma_{m_{QS}}}{A_{arable}}\right)^2 + \left(\frac{\sigma_{m_P}}{A_{arable}}\right)^2 + \left(\frac{(m_{GW}+m_{QS}-m_P)}{A_{arable}^2} \times \sigma_{A_{arable}}\right)^2} \tag{H8}$$

## 9 Error of the ET value calculated from TFA concentrations

The error of the $ET$ values from TFA concentrations as calculated in Eq. (F9) results from the errors $\sigma$ in annual discharge $Q_S$
and groundwater discharge $Q_{GW}$ and the volume-weighted TFA concentrations in precipitation $c_P$, discharge $c_{dis}$, and groundwater $c_{GW}$ as shown in Eq. (H9):

$$\sigma_{ET} = \sqrt{\left(\left(\frac{c_{dis}}{c_P} - 1\right) \times \sigma_{Q_S}\right)^2 + \left(Q_S \times \frac{1}{c_P} \times \sigma_{c_{dis}}\right)^2 + \left(\left(-Q_S \times \frac{c_{dis}}{c_{P2}} - Q_{GW} \times \frac{c_{GW}}{c_P^2}\right) \times \sigma_{c_P}\right)^2 + \left(\frac{c_{GW}}{c_P} \times \sigma_{Q_{GW}}\right)^2 + \left(Q_{GW} \times \frac{1}{c_P} \times \sigma_{c_{GW}}\right)^2} \tag{H9}$$



**Code availability**

An R-script, performing the necessary calculations for this paper's results, can be found at 10.5281/zenodo.15673020.

**Data Availability**

Data necessary to reproduce this paper's results are supplied under 10.5281/zenodo.15673020.

**Author Contribution**

Conceptualization: IF, JL

Data curation: IF

Formal analysis: IF

Funding acquisition: JL, JL

Investigation: IF, DN

Methodology: DN, IF, FF

Project administration: JL, MM

Resources: JL, MM

Software: IF

Supervision: JL, MM

Validation: IF

Visualization: IF

Writing (original draft preparation): IF, JL

Writing (review and editing): IF, JL, MM, FF, DN


All authors have read and agreed to the submitted version of the paper.

**Competing interests**

The authors declare that they have no conflict of interest.


**Acknowledgments**

We want to express our sincere thanks to the individuals and organizations who contributed to this research. We thank Badenova and Simon Brenner for providing groundwater samples essential for our analysis. Our appreciation also goes to the anonymous engineering office for supplying conductivity values for the Dreisam aquifer. We acknowledge the Deutscher



Wetterdienst (DWD) and Monica Rauthe's assistance in providing the corrected HYRAS version. Special thanks to the Regierungspräsidium Freiburg and Stefanie Rauscher for supplying discharge data for the Brugga River. We are grateful to the AK Andexer for granting us access to the LC-MS system and allowing us to conduct necessary chemical analyses. Furthermore, we thank Adelheid Nagel for her help in the LC-MS sample preparation. Additionally, we would like to thank Barbara Herbstritt for her facilitation of ion chromatography and Heinel Franziska for her sampling efforts. Finally, we would

like to thank all partners and founders of the Interreg Project Reactive City for their support in making this work possible. Without the contributions of these individuals and organizations, this study would not have been possible.

Furthermore, we declare that Grammarly AI was used to improve the readability, grammar, and spelling of this manuscript, and GitHub Copilot supported coding.

**Financial support**

This research was co-funded by the European Union within the Interreg Reactive City Project.

**Disclaimer**

Publisher's note: Copernicus Publications remains neutral with regard to jurisdictional claims made in the text, published maps, institutional affiliations, or any other geographical representation in this paper. While Copernicus Publications makes

every effort to include appropriate place names, the final responsibility lies with the authors.

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
