# Peer review of "Diffuse sources of TFA: atmospheric and terrestrial inputs, retention and pathways at the catchment scale"

_EGUsphere, 2025_

## Referee Comment (RC2)

The paper studies the retention and transport of Trifluoroacetate (TFA) in three catchments – two where atmospheric deposition is the only source and one with additional sources of agriculture and waste water treatment plants – and a spring in the southern Black Forest of Germany. The objectives of this paper include 1) investigating whether TFA retentions occurs within these catchments; 2) estimating TFA input through agricultural activities based on water flux measurements and their TFA concentrations; 3) identifying the primary pathways through which TFA reaches the stream; and 4) estimating evapotranspiration (ET) using TFA as a tracer. The paper uses the TFA concentration data in precipitation, stream water samples and groundwater and the fluxes of these water flows to calculate the mass balance of TFA at annual scale to examine whether TFA retention occurred in these catchments. They concluded that retention does not occur in catchments with only atmospheric deposition. Considering "non-retention" to hold for bigger downstream Dreisam catchment as well, the agricultural surplus there was back calculated from TFA mass balance. Based on the "non-retention" in upstream catchments, their evapotranspiration was estimated using TFA as tracer. The subsurface storm flow was identified as primary pathway for mobilizing TFA to streams by examining the temporal dynamics of TFA and their correlation with other solutes.

TFA, one of the Per- and Polyfluoroalkyl Substances (PFAS) or "forever chemicals", has been increasing concentrations across various environmental compartments, raising concerns about their potential ecological and health implications. This study on the retention and mobilization of PFA in agricultural and forest catchments provides valuable insights into catchment scale PFA dynamics and its potential as a tracer to hydrologists. The paper is well written and concise. It will be of interest to the wider audience of HESS. However, there is need to provide more justification for underlying assumptions, acknowledge limitations of a 2-year study and present the results and conclusions in more cautious language. I recommend publishing the manuscript following major revisions.

**Major comments:**

*Introduction*

The introduction is concise and frames the knowledge gap effectively. I appreciate how the research questions are laid out clearly. I would suggest stating in introduction that TFA is a PFAS (Per- and Polyfluoroalkyl Substances) as PFAS are more commonly known and may help attract broader interest.

*Methods*

Lines 94-96 suggest event water reaches stream mainly as overland flow, whereas lines 104-110 suggest event water flows mainly through subsurface flow pathways. Please clarify whether overland flow occurs substantially during storms in this catchment or if it occurs only in localized areas with most event water reaching stream through subsurface flow pathways.

The following sentences are vague with lack of logical flow from first sentence to second. Please edit them.

Line 105-107: "During discharge peaks, SSF contributed up to 50% of the streamflow in a catchment in the vicinity of the DRC (Bachmair and Weiler, 2014). Then, a network of pores interconnects to a system of hydrological flow paths that respond to precipitation according to its connectivity. Thus, SSF intensity depends both on soil moisture and event magnitude."

Since SSF refers to a flow rather than the flow pathways through which water moves, I suggest modifying the following sentence accordingly.

Line 107-109: "Although event water may travel rapidly through SSF, a considerable portion of the mobilized water consists of "old" water already present in the flow system"

Line 115: I suggest adding a brief sentence explaining that these precursors are "volatile fluorinated organic compounds" and are typically used as refrigerant to provide additional context to readers.

The following sentence is a bit dense. It might be better to split it into two sentences and make the meaning more explicit. Also, while it's common to say that samples are enriched, it's less usual to say concentrations are enriched—maybe using 'concentrations are higher' would sound more natural."

Line 116-117: "Concentrations are typically enriched in samples from low precipitation volumes, necessitating precipitation volume weighting for representative input concentrations

Lines 121 onwards: Since WWTP are also terrestrial sources, I suggest replacing "terrestrial sources" with "agricultural activities".

Lines 123-125-> I recommend introducing pesticides in the first sentence itself. For example, "Pesticides are fluorinated compounds that contain …." can naturally lead into second sentence.

Line 137 -> Since ski wax is mentioned here, please clarify whether ski wax is an additional source for TFA in this catchment or can be disregarded.

Line 149 -> It seems like drinking water TFA level increased from 0.64 μg/l to 0.84 μg/l from November 2024 to April 2025, amounting to ~30% increase. This hints at seasonality in aquifer concentrations. Was this seasonality considered when calculating mass flux of TFA? In other words, was the groundwater discharge weighted with a "constant" groundwater TFA concentration or seasonal values of groundwater TFA concentration?

Line 152 -> Table 4 is introduced before Table 1. Please number the tables in the order they are first mentioned in the manuscript.

Line 161-164 -> This sentence is quite long and complex. It can be either split or restructured to improve readability

Table 1 appears in the manuscript but not referenced in the main text. Please ensure all tables are referenced in the manuscript.

Lines 200 -> How large and frequent were these data gaps? Please provide a clear justification for using linear interpolation from weekly samples for gap filling a time series with a 15- or 10-minute resolution. Also, what was the source of discharge for Talbach? I suggest adding the years for which you have data for each of these catchments as well here.

Line 213-214 -> Over how many years was Mann-Kendall trend calculated? While change in storage is often neglected in long term water balance, this term cannot be neglected at annual scale. Please add a stronger justification as to why this term was ignored for annual water balance. Were the groundwater levels at the start and end of time period same perhaps?

Why was the water input into Zipfeldbel spring considered as precipitation occurring in nine pixels around the spring? The aquifer feeding it could be receiving water from a broader area which could even be located far away. Therefore, the input should include precipitation over the recharge area. Has the recharge zone for the aquifer been delineated?

There could be subsurface flows occurring within the aquifer feeding the spring. If such flows are present, they should be accounted for in the water balance. Please clarify whether the aquifer has known inflows or outflows or even other springs. If they are present but excluded, please provide a justification for neglecting them in the water balance.

What is the water transit time for the aquifer? What is the justification for considering the yearly change in storage term as zero? Possible reasons could be very long transit times, muti year stable groundwater levels or spring discharge, same groundwater level at the start and end of year, etc.

Line 260: Can you include the formula for calculating $c_{dis}$ and $c_{GW}$? How is $c_{QGW}$ different from $c_{GW}$?

**Results**

Line 276 -> It looks like the highest TFA levels in Dreisam river occur in May 2023. Also, it looks to me that precipitation was not that low during the winter. I suggest putting numbers to support these statements, since it is hard to judge them visually from the figure 2. Or perhaps the intention was to highlight the high levels of TFA in river despite low levels of TFA in precipitation during winter. In that case, please rephrase the sentence accordingly.

Line 306 -> Correlations for Deuterium excess is missing in Table 2.

Line 315-> If 2023 as dry year and 2024 as wet year, then water balance for years 2023 or 2024 separately wouldn't be close without a change in storage term.

Table 5 -> There is no equation 9 in the manuscript. Maybe it was accidently excluded.

While it is reasonable to make assumptions like the WWTP TFA levels remained constant over a year, I still suggest explicitly stating somewhere that the TFA levels were based on a single day measurement.

While groundwater TFA levels seem constant over two years, observations over longer period might reveal a trend. Considering that anthropogenic sources have bene increasing, most probably the groundwater TFA levels have also been increasing here. Please explicitly acknowledge this so that it doesn't give the impression that the groundwater TFA levels have stabilized.

Considering that groundwater TFA levels could have been seasonal, was this accounted for in calculating mass flux?

**Discussion**

The hypothesis on organic soil zone as temporal TFA storage which contributes to TFA pulses in streams during storm flows is interesting and insightful. This can be a good framework to explore TFA dynamics in other catchments as well.

While the short-term mass balance suggests limited TFA retention, I would caution that a two-year dataset may not be sufficient to conclusively rule out retention processes. Given the known

groundwater residence times in the Brugga catchment (ranging from a few to over ten years), some of the TFA currently reaching the stream could be originally from legacy sources, while recent TFA inputs may still be retained in subsurface. As such, I would suggest interpreting the apparent balance with some caution, and perhaps acknowledging the potential for longer-term storage and delayed transport within the system. I also suggest adding some discussion about systems with deep groundwater and long transit times where TFA might be retained more.

I appreciate the acknowledgement of potential uncertainties introduced by the use of data from a single precipitation sampling station as well as the spatial variability of TFA concentrations in precipitation.

There is a TFA surplus in Dreisam catchment which you attribute to manures. However, could this surplus be from legacy storage of TFA from previous years? Given that 2023 was dry year, the TFA from previous years could have been retained in catchment and subsequently be mobilized in the wet year, 2024.

The ET estimates for each year from water balance as well as from TFA mass balance rely on the assumption of closed water balance for each year. Therefore, I suggest including direct ET measurements, if they are available, or summarizing results of previous studies that report ET ranges for these catchments to support that these are reasonable estimates.

TFA could be a valuable tool as tracer, especially considering that it is less expensive to measure than isotopes. Highlighting this could strengthen the case for its use as tracer.

*Conclusion*

Line 470 -> This is a strong statement. While the hypothesis that organic soil zone is the primary TFA storage, and SSF is the primary mechanism by which it reaches stream has high chances of being true, we need additional data like soil TFA profile or isotope tracer studies to support this. Therefore, I advise you to rephrase this into a more cautious statement and acknowledge the need for direct measurements.

In general, I suggest a more cautious wording of conclusions to reflect the limitations and assumptions of the study. That said, these assumptions and limitations do not reduce the value of your work.

**Minor comments:**

Line 54 -> "also" should be deleted.

Line 56-57 -> "we took weekly sample of precipitation at a weather station and stream water in three nested catchments and a hillslope spring.

Line 60 -> Consider changing to "headwaters, which are free of arable land."

Line 84 -> This sentence is not clear and grammatically doesn't make sense.

Line 86 -> Move inclinations to earlier: "with inclinations up to 62°"

Figure 1 -> Consider making the background of labels on figure transparent or removing them from figure, so that river network and catchment boundaries are visible.

Line 95-96 -> Since abbreviations "SOF" and "HOF" are not used even once after their introduction, please remove them

Line 121 -> For consistency, use "Fig. 1"

Line 122 -> Consider using "released from" instead for better flow

Line 128 -> The value of "n" does not add much meaning here and could be removed, unless you want to point out that the small number of samples makes this estimate uncertain. If so, it would be good to explicitly mention it.

Line 134 -> The full form of WWTP has already been introduced and therefore can be skipped here

Line 142 -> Did you probably mean "river"?

Line 167-170 -> Consider enclosing A and B – abbreviations for eluents – in brackets or quotes or put them in italics to avoid confusion

Line 175 -> remove brackets around Synek, 2008

Figure 2 -> Please correct the unit for Q in panel b. The green time series for spring looks like a solid line for most of the time except a short time period July-Oct 2024, when it is dotted line.

 Line 279 -> Consider specifying the months you mean by "late summer"?

Line 372 -> "On" should be used instead of "At".

Line 454 -> "Eq 9" is missing

---

## Referee Comment (RC3)

This study presents a valuable and comprehensive investigation into the atmospheric and terrestrial pathways, retention, and export of trifluoroacetate (TFA) at the catchment scale. The two-year dataset and multi-compartment sampling approach (precipitation, streamflow, springs, WWTPs) provide an important basis for understanding TFA dynamics. The manuscript is generally well written and organized; however, several methodological clarifications and interpretations should be addressed before publication.

**Line 160–165:**

Please clarify the rationale for choosing this separation column for TFA analysis. What are its advantages compared with other commonly used columns for PFAS analysis, such as the Hypersil Gold C18 column?

**Line 165–170:**

The role of 50 mM ammonium hydrogen carbonate in pure water as mobile phase A should be clarified. If the separation column is hydrophilic, the use of methanol as mobile phase B may not be ideal. Please confirm the column chemistry and justify the chosen mobile phase composition.

**Line 175–176:**

You mention Mill-Q blank samples as procedural blanks. Were these blanks also used to assess potential TFA contamination from the LC–MS system (e.g., tubing, fittings, internal components)? Please describe any specific cleaning or pre-conditioning procedures used to minimize TFA background signals from the instrument.

**Line 190-195:**

It is mentioned that the same separation and guard columns used for LC–MS analysis were also used for IC analysis of major anions and cations. Does this imply that the IC system could potentially be used for TFA determination as well? If so, was this tested or verified?

Additionally, please clarify whether "supplier" refers to the supplier of the IC instrument or of the columns. Did you determine quantification limits for each ion using your own calibration curves under actual operating conditions, which might be more accurate than supplier-provided values?

**Line 275–276:**

Please elaborate on the factors leading to the highest TFA levels in the Dreisam River during the 2023–2024 winter.

**Figure 2:**

Does the gray shading in panels (a–c) represent dry and wet conditions? Please specify this in the figure caption.

**Line 303–304:**

Based on the correlation data in Table 2, it seems that TFA showed positive correlations with all tracers except nitrate. Please confirm and revise accordingly.

**Line 306:**

The sentence "The same was true for the negative correlation with deuterium excess" is ambiguous. Please rephrase for clarity (e.g., "Similarly, TFA exhibited a negative correlation with deuterium excess").

**Line 364–365:**

You state that "Potassium negatively correlated with TFA; however, concentrations were below LOQ and could not be reliably interpreted." How was the correlation established if potassium concentrations were not quantifiable? Please clarify or reconsider this statement.

**Line 366–367:**

Given that the spring pH is around 6, the previously cited finding that "TFA sorption to soils decreased with increasing pH up to pH 5" (Richey et al., 1997) may not adequately explain the observed behavior at the Zipfeldobel spring. Please discuss this limitation.

**Line 428–430:**

Considering TFA's high mobility, the lack of significant retention in water bodies contrasts with reported TFA retention in plants and soils (Likens et al., 1997; Berger et al., 1997). Please elaborate on possible mechanisms or environmental conditions explaining this discrepancy.

**Line 430:**

The sentence "Potentially, differences originate from the study design of both field experiments" should include more detail on what specific design differences (e.g., sampling frequency, soil types, hydrological setting) might explain the divergent results.

**Line 447–449:**

Please provide references supporting the statement that patterns/concentrations "were attributed to the distribution of TFA precursor molecules in the atmosphere."

**Line 380 vs. Line 470:**

You stated that "our hypothesis of a temporal TFA storage, most likely associated with organic soil, seems valid," yet later conclude that "We identified the organic soil zone as a primary TFA storage." Since the data suggest only temporary accumulation, the latter conclusion may overstate the findings. Please rephrase to maintain consistency and avoid overinterpretation.

**Line 475:**

You compare TFA loads from farming activities with values reported for precursor PPP degradation in Joerss et al. (2024). If those values were derived from a different catchment, the comparison may not be meaningful. Please clarify whether the data are directly comparable.

---

## Author Comment (AC1)

**EGUSPHERE-2025-2882, reply on RC1**

Comments are copied in plain text.

Replies are blue.

Changes to the manuscript are in green.

**General comments:**

The authors made a comprehensive and interesting study on the size and applicability of TFA concentration on catchment scale. The study includes many data that may be used by others interested in similar catchment scale studies or in other study types. In general discussions and conclusions are both interesting and justified. However, I am skeptical about the justification of the calculations on agricultural input. While the higher TFA concentrations at the Dreisam Gauge is likely to be caused by an input from agricultural activity, as the authors suggest, I think the numbers used in the calculations in combination with the general understanding of agricultural sources are so uncertain that the calculated mass of TFA from agriculture makes no sense and should be removed from the abstract and probably from the article and more emphasis should be put on the need for more research into this important aspect of the study. Please see specific comments for more details on this.

We thank the reviewer for the comprehensive feedback and the time invested in the review. We agree that calculating an agricultural TFA excess is highly uncertain. We followed the reviewer's suggestion and removed the TFA surplus from the abstract and the article.

**Specific comments:**

Line 51 "the lack of transformation rates". Not only transformation rates are lacking but for most potential TFA-pesticides also whether the pesticides are transformed into TFA at all.

To take the potential of "non-transformation" into account, we changed the sentence to:

Moreover, it remains unknown whether, and to what extent, the degradation of PFAS PPPs releases TFA.

Line 93: How were HFCs (and which) used as tracers? Normally you would expect CFCs to be used as tracers. Are HFC/CFC data in Uhlenbrook et al, I cannot find them?

We assume the existence of two Uhlenbrook 2002 references has led to some confusion. There is a paper called "Hydrograph separations in a mesoscale mountainous basin at event and seasonal timescales" with two more co-authors cited as "Uhlenbrook et al. 2002" and another one called "Process-oriented catchment modelling and multiple-response validation" with one more coauthor cited as "Uhlenbrook and Leibundgut, 2002". In short: The freons F-11, F-12 and F-113 were used and the full article can be found under doi: 1029/2001WR000938. And as the reviewer stated: Those molecules also contain chlorine atoms and are therefore called chlorofluorocarbons (CFCs), we changed the statement in brackets accordingly:

(major ions, silica, and clorofluorocarbons – CFCs, namely freons F-11, F-12 and F-113)

Line 95-96: "HOF" and "SOF" are these abbreviations necessary - they are not used any further?

We introduce the abbreviations in the new version of the draft and use them in the paragraph.

Hortonian Overland Flow (HOF) from impervious surfaces, such as roads, rock outcrops, or urban areas, and by Saturation Overland Flow (SOF) from saturated areas, including wetlands and riparian zones.

In terms of magnitude, SSF is the dominating process of storm discharge generation when compared to HOF and SOF (Steinbrich et al. 2016).

Line 143: "a polyethylene (PE) tank located below the funnel". This is not much information on the precipitation collection. What material was the funnel made of? Did you test that it did not leach or adsorb TFA? Was it assured that there was not evaporation from the tank? Was it a constantly open funnel, so that also some of the dry deposition would be sampled? In general, dry deposition is not considered in the study, but some studies claim that it is a substantial fraction of atmospheric TFA input (most extreme case by Zhuang et al

(https://www.sciencedirect.com/science/article/abs/pii/S0304389424009622), and most studies estimate at least some dry deposition. I guess there will also be quite some fog/mist in the area, that can be captured by especially conifers, but will probably not be captured by the precipitation sampler, or?

We revised the section to address the reviewers' questions regarding blank testing of materials and evaporation protection. We follow the author's suggestion and added dry deposition as a potential source of errors to our discussion. We propose the following changes in the manuscript:

On the same day, we retrieved precipitation samples: rainwater was stored in a tank located below the funnel, which was emptied after sampling. The tank, funnel, and connecting tube (0.82 m) were composed of polyethylene (PE), and the material was blank tested to exclude any contamination of the material. Because of the strong hydrophily of the TFA molecule we did not consider sorption to the hydrophobic PE-material. The setup allowed the tank to be placed in the high grass to keep temperatures low. Furthermore, the tank was protected by a black rubber mat to minimize evaporation, shielding it from solar radiation. Evaporation protection was necessary because the funnel was left permanently open to allow for additional dry deposition sampling.

Although our precipitation sampler was permanently open, dry TFA deposition may be enhanced in forested environments due to the larger surface area provided by tree canopy, which increases the potential for particle deposition. Indeed, studies have reported increased TFA levels in throughfall compared to bulk precipitation, supporting the likelihood of additional inputs via fog and dry deposition in forest ecosystems (Jordan und Frank 1999). This suggests that our atmospheric TFA input estimates based solely on open field bulk precipitation measurements might be too low, resulting in an apparent lack of retention when performing mass balance calculations.

Line 147: "Storage time was up to four months". Stored how?

We added information on sample storage to the sentence.

Samples were stored in the dark and at room-temperature for up to for month, but...

Line 275: "the highest TFA levels in the Dreisam River were observed during the 2023/2024 winter". To me this is not clear from the figure - it looks relatively constant, when considering the fluctuation from sample to sample...?

The intention was to emphasize the high TFA levels in the Dreisam River, despite the low precipitation levels. We rephrased accordingly.

2. Despite low TFA levels in precipitation, high TFA levels were observed in the Dreisam River during winter.

Line 277: "TFA concentrations increased with discharge". Is this statistically significant - it is not so easy to see from the figure, I think. Sometimes there seem to be a clear positive relationship between the two, sometimes not...

We added a sentence to link the information on significance levels and correlations to this observation.

We also refer to Table 2 where correlations and significance levels are listed. The discharge-TFA correlation is statistically significant at Dreisam and Brugga (p≤0.001), but not at the Spring Zipfeldobel.

Line 334: "The Dreisam River exported  $48 \pm 21$  % more TFA." This seems like a very uncertain calculation, since according to the figure in 2023 there was no surplus and in 2024 the surplus was around 100%? Taking average of two such different numbers makes no sense to me. Furthermore, how can the uncertainty be so small when taking average of two so different years? There should be input from agriculture in both years unless there is a very good (hydrological) reason for this to be so different. Although quantifying the contribution from agriculture is very important, I am not convinced it is scientifically reasonable to do from these data as apparently uncertainty is very high (which is actually expected for a whole catchment exercise)

We share the concerns of the reviewer regarding the average years uncertainty. We removed the "average" year from the plot and replaced it with the cumulative input and output over the two years.

Line 341: "main Dreisam catchment" is this the same as what is denoted DRC throughout the text?

We forgot to use the abbreviation; we changed to DRC.

Figure 4: Why use standard error, not standard deviation, so that it is easier to compare statistical significance?

As the reviewer suggested, the SE is better suited to compare significance. If we want to make a statement about retention/excess, then we need a threshold when defining these differences. The SE provides error bands suited to do so.

**We added this to the draft:**

To be able to compare statistical significance, whiskers indicate standard errors, calculated from bootstrapping standard errors for mean concentrations and volumes, and consecutive Gaussian error propagation.

Table 5: "Eq. 9" Do you mean Eq.8? I see no eq. 9.

We agree with the reviewer: this was a spelling mistake; it should be Eq. 8. We have made the correction accordingly.

Line 382: In general, water isotopes seem to show minor seasonal fluctuation, compared to what you would expect in the precipitation (though precipitation isotopes for the present study seems not to be

shown). It guess this would suggest the rivers are mainly fed by groundwater? Does this fit with your understanding of the system? Does it fit with the fluctuations in TFA? Do you have stable isotope data for precipitation that could be included?

The rivers are mainly groundwater-fed. We calculated Kirchner's new water fraction (the proportion of water younger than two month) from water stable isotopes and ended up with a low single-digit percent value. Still under storm conditions, SSF can make up to 50 % of the discharge (Bachmair and Weiler, 2014). For precipitation stable isotope data we refer to Figure F2.

**We included a paragraph in the draft to account for the reviewer's comment:**

Generally, this aligns with our understanding of a groundwater-dominated system. We calculated Kirchner's new water fraction, i.e. the proportion of water younger than two months (Kirchner 2019) from water stable isotopes and ended up with a low single-digit percent value. Under dry conditions, groundwater influence stabilized TFA concentrations in streamflow and under storm conditions SSF gains importance, mobilizing TFA.

Section 4.2: Although highly interesting, this section seems very speculative and I think it should be minimized and uncertainties put more forward.

Another reviewer also expressed concern about the lack of a thorough discussion of uncertainty. We rephrased the section to focus on adding a more thorough discussion of uncertainties. We refer to reviewer 2's comments on the discussion section.

Line 421-427: Taking an average of two very different years (in terms of hydrology and TFA balance) doesn't make sense to me even if the average precipitation volume of the area lies between the two.

We share the concerns of the reviewer and have removed the calculation of a mass balance for the average year. Instead, we show the TFA import and export over the two years. Still, the average year seems necessary for calculating ET; therefore, we further stress the uncertainty, which is reported in the results coming with this method in Section 4.3.

Calculating *ET* based on weighted mean TFA concentrations resulted in higher error margins than using the deficit between water import and export. High fluctuations in TFA precipitation concentrations are the primary cause of elevated uncertainty.

Line 444: 100% molar yield is highly unlikely (pesticides are rarely degraded 100% to one compound, and a significant fraction of TFA and possibly the pesticides themselves will be removed with the crops). It is important to mention that this is highly unlikely, and also important to mention that so far, TFA from PPP is mainly hypothetical/potential (Joerss et al use the term "estimations of TFA"

formation potentials "). For the few compounds, where TFA was demonstrated in the EFSA conclusions and elsewhere, (much) smaller fractions than 100% was found.

The reviewer is correct in stating the lack of data on PPP transformation to TFA; therefore, we have added a section that takes this into account.

Joerss et al. (2024) published a dataset estimating the potential agricultural TFA input based on a European dataset regarding emissions from plant protection products; however, it remains unknown whether and to what extent PPP degrade to TFA. The TFA excess in the DRC suggest at least some additional agricultural input.

Line 445: I doubt that manure will add much TFA compared to pesticides, but I guess you could calculate from published values for TFA in manure, mentioned in the Introduction?

We appreciate the idea and estimated liquid manure input, indeed it seems negligible:

Assuming a TFA concentration of 100 µg L-1 in liquid manure, as reported by the German Environment Agency (2023), an application rate of 15 tonnes per hectare (t ha-1) would result in an annual input of less than 1 kg of TFA across the entire agricultural area of the DRC (5.2 km2). Consequently, liquid manure can be considered a negligible source of TFA input in this context.

Line 454: "Eq. 9" Do you mean Eq.8? I see is no eq. 9.

This is a mistake, it is Eq. 8.

Line 458-465: What about the year-to-year variation that may be quite large, as you show in table 4 and which has also been shown by others (much higher than the 5% you mention as an average yearly increase). Maybe this should be discussed here as well?

We agree with the reviewer and have changed the section accordingly.

Adding to the uncertainty, the interannual variability of TFA concentrations in precipitation can be considerable (see Table 4 and Jordan und Frank (1999) and Henne et al. (2025)), as reflected in the relatively large error margins for *ET* values from TFA, ranging from 45% at Brugga to 70% at Zipfeldobel. This constitutes a methodological challenge, as the sampling period must be sufficiently long to capture interannual variation, while be short enough to minimize errors arising from the progressive increase in TFA concentrations over time.

Also see the change regarding the reviewer's comment on line 478.

Line 475: "reported for the degradation of precursor PPP". I would suggest to change into "reported for the potential degradation of precursor PPP"

We changed the sentence as the reviewer suggested.

Line 478: "mean residence times of 2-5 years". Why not shorter than 2 years? Is it not applicable for one year (11-16 months, line 460) as in the present manuscript?

**We agree with the reviewer and changed the sentence to:**

This approach is applicable in catchments with solely atmospheric TFA input. Mean residence times should remain below 5 years minimize the error from rising TFA concentrations in precipitation. Main uncertainties arise from difficulties in finding representative precipitation TFA concentrations, as overall TFA concentrations are increasing, and the interannual variability can be quite high.

---

## Author Comment (AC2)

**EGUSPHERE-2025-2882, reply on RC2**

Comments are copied in plain text.

Replies are blue.

Changes to the manuscript are in green.

The paper studies the retention and transport of Trifluoroacetate (TFA) in three catchments – two where atmospheric deposition is the only source and one with additional sources of agriculture and waste water treatment plants – and a spring in the southern Black Forest of Germany. The objectives of this paper include 1) investigating whether TFA retentions occurs within these catchments; 2) estimating TFA input through agricultural activities based on water flux measurements and their TFA concentrations; 3) identifying the primary pathways through which TFA reaches the stream; and 4) estimating evapotranspiration (ET) using TFA as a tracer. The paper uses the TFA concentration data in precipitation, stream water samples and groundwater and the fluxes of these water flows to calculate the mass balance of TFA at annual scale to examine whether TFA retention occurred in these catchments. They concluded that retention does not occur in catchments with only atmospheric deposition. Considering "non-retention" to hold for bigger downstream Dreisam catchment as well, the agricultural surplus there was back calculated from TFA mass balance. Based on the "non- retention" in upstream catchments, their evapotranspiration was estimated using TFA as tracer. The subsurface storm flow was identified as primary pathway for mobilizing TFA to streams by examining the temporal dynamics of TFA and their correlation with other solutes. TFA, one of the Per- and Polyfluoroalkyl Substances (PFAS) or "forever chemicals", has been increasing concentrations across various environmental compartments, raising concerns about their potential ecological and health implications. This study on the retention and mobilization of PFA in agricultural and forest catchments provides valuable insights into catchment scale PFA dynamics and its potential as a tracer to hydrologists. The paper is well written and concise. It will be of interest to the wider audience of HESS. However, there is need to provide more justification for underlying assumptions, acknowledge limitations of a 2-year study and present the results and conclusions in more cautious language. I recommend publishing the manuscript following major revisions.

We are grateful for the reviewer's time and effort invested in providing the thorough review. In the following, we aim to integrate the comments by justifying the assumptions we made, acknowledging the limitations arising from the two-year sampling period, and presenting the results more carefully.

**Major comments: Introduction**

The introduction is concise and frames the knowledge gap effectively. I appreciate how the research questions are laid out clearly. I would suggest stating in introduction that TFA is a PFAS (Per- and Polyfluoroalkyl Substances) as PFAS are more commonly known and may help attract broader interest.

We agree with the author's opinion. We modified the first sentence accordingly.

Trifluoroacetate (TFA) is the smallest member of the Per- and Polyfluorinated Substances (PFASs) group and a degradation product of various anthropogenic fluorinated compounds.

**Methods**

Lines 94-96 suggest event water reaches stream mainly as overland flow, whereas lines 104-110 suggest event water flows mainly through subsurface flow pathways. Please clarify whether overland flow occurs substantially during storms in this catchment or if it occurs only in localized areas with most event water reaching stream through subsurface flow pathways.

Acknowledging the need for further clarification, we rephrased the section and added a statement of the magnitudes of SOF, HOF, and SSF.

The first component is event water, which may account for up to 50% of streamflow during storm events. It is generated by Hortonian Overland Flow (HOF) from impervious surfaces, such as roads, rock outcrops, or urban areas, and by Saturation Overland Flow (SOF) from saturated areas, including wetlands and riparian zones. Furthermore, Subsurface Storm Flow (SSF), which was observed in the steep hillslopes, also carries an event water component (Bachmair und Weiler 2014). SSF occurs when a network of pores interconnects to a system of hydrological flow paths that respond to precipitation according to its connectivity. Connectivity is dependent on soil moisture, and therefore, the intensity of SSF is dependent on both pre-event conditions and the magnitude of the event. Although event water may travel rapidly through SSF, a considerable portion of the mobilized water consists of "old" water already present in the flow system (Kienzler und Naef 2008). In terms of magnitude, SSF is

the dominating process of discharge generation when compared to HOF and SOF (Steinbrich et al. 2016).

The following sentences are vague with lack of logical flow from first sentence to second. Please edit them.

Line 105-107: "During discharge peaks, SSF contributed up to 50% of the streamflow in a catchment in the vicinity of the DRC (Bachmair and Weiler, 2014). Then, a network of pores interconnects to a system of hydrological flow paths that respond to precipitation according to its connectivity. Thus, SSF intensity depends both on soil moisture and event magnitude."

Since SSF refers to a flow rather than the flow pathways through which water moves, I suggest modifying the following sentence accordingly.

Line 107-109: "Although event water may travel rapidly through SSF, a considerable portion of the mobilized water consists of "old" water already present in the flow system"

We agree and modified the sentences as suggested by the reviewer. For details, see the modifications in the section from the comment above (regarding Line 94).

Line 115: I suggest adding a brief sentence explaining that these precursors are "volatile fluorinated organic compounds" and are typically used as refrigerants to provide additional context to readers.

Adding a statement on why and how these gases are actually used adds valuable context, as the reviewer suggested. We modified the sentence accordingly:

Atmospheric TFA is a transformation product of volatile organic precursor molecules such as HFC-134a and HFO-1234yf, and reaches the erth's surface through wet deposition (Freeling et al. 2020; Franklin 1993). Those gases are typically used as refrigerants, primarily in automobile air conditioning systems.

The following sentence is a bit dense. It might be better to split it into two sentences and make the meaning more explicit. Also, while it's common to say that samples are enriched, it's less usual to say concentrations are enriched—maybe using 'concentrations are higher' would sound more natural."

Line 116-117: "Concentrations are typically enriched in samples from low precipitation volumes, necessitating precipitation volume weighting for representative input concentrations

**We changed the sentence to make it less dense and improve the wording:**

TFA concentrations are higher in samples from smaller precipitation events. Therefore, volume weighting is necessary to obtain representative mean precipitation concentrations.

Lines 121 onwards: Since WWTP are also terrestrial sources, I suggest replacing "terrestrial sources" with "agricultural activities".

**We replaced terrestrial sources with agricultural activities:**

There, in addition to the ubiquitous atmospheric input, TFA can additionally be emitted from agricultural activities.

Lines 123-125-> I recommend introducing pesticides in the first sentence itself. For example, "Pesticides are fluorinated compounds that contain ...." can naturally lead into second sentence.

**We appreciate and follow the suggestion:**

Pesticides are fluorinated compounds that may contain carbon-bound –CF3 groups and potentially degrade to TFA (Scheurer et al. 2017; Sun et al. 2020). The surge in the use of such pesticides (Ogawa et al. 2020; Alexandrino et al. 2022) could result in high TFA releases from agricultural areas.

Line 137 -> Since ski wax is mentioned here, please clarify whether ski wax is an additional source for TFA in this catchment or can be disregarded.

Because we found no further literature regarding the topic, it remains unclear whether PFAS in ski wax contribute to TFA pollution. Ski wax contains longer-chain PFAS (C4-C14), and since those substances are very persistent, degradation in the short term is unlikely. If they had contributed much, we should have seen TFA peaks in Brugga but not Talbach during snowmelt, which we could not detect. Therefore, we removed the ski wax from the draft.

Line 149 -> It seems like drinking water TFA level increased from 0.64  $\mu$ g/l to 0.84  $\mu$ g/l from November 2024 to April 2025, amounting to ~30% increase. This hints at seasonality in aquifer concentrations. Was this seasonality considered when calculating mass flux of TFA?

In other words, was the groundwater discharge weighted with a "constant" groundwater TFA concentration or seasonal values of groundwater TFA concentration?

The seasonality was not considered. To take the comment into account, we increased the uncertainty in TFA groundwater concentrations to 30%. Therefore, the fluctuations should be reflected in the results.

**In appendix H:**

The mean TFA groundwater concentration error was set to 30% because measurements in the Freiburg drinking water supply indicated seasonal variability, with the same magnitude.

**And in the manuscript:**

The comparison suggested that groundwater-TFA levels remained relatively constant over the two-year study period but might exhibit seasonal variability (2024-January:  $0.81~\mu g~L^{-1}$ , 2024-November:  $0.64~\mu g~L^{-1}$ , 2025-April:  $0.84~\mu g~L^{-1}$ ). To account for seasonality, we set the error in the groundwater TFA concentration to 30% to ensure this uncertainty is reflected in the results.

Line 152 -> Table 4 is introduced before Table 1. Please number the tables in the order they are first mentioned in the manuscript.

We apologize for the formal error. Since we believe it is impractical to have the last table in the results section numbered Table 1, we have now integrated the values directly into the text to improve readability and adhere to the formatting guidelines.

(See comment above)

Line 161-164 -> This sentence is quite long and complex. It can be either split or restructured to improve readability

**We changed the sentence to improve readability:**

Ion exchange liquid chromatography-electrospray tandem mass spectrometry (LC-ESI MS/MS) analysis was performed using a Shimadzu LC-AD20 coupled with API 5500 Q Trap triple-quadrupole mass spectrometer (Applied Biosystems/MDS Sciex Instruments, Concord, ON, Canada). The electrospray interface operated in negative ionization mode (see Table 1 for MS conditions).

Table 1 appears in the manuscript but not referenced in the main text. Please ensure all tables are referenced in the manuscript.

We apologize for the formal error. The Table is now correctly referenced.

(See comment above)

Lines 200 -> How large and frequent were these data gaps? Please provide a clear justification for using linear interpolation from weekly samples for gap filling a time series with a 15- or 10-minute resolution. Also, what was the source of discharge for Talbach? I suggest adding the years for which you have data for each of these catchments as well here.

We included the percentage of missing flow meter readings. We justify interpolating weekly manual measurements on a 15-minute timeseries, which improves accuracy when calculating annual mean discharges in years with high flow meter downtime. Historical Talbach discharge values were obtained from LUBW. In the updated version of the draft, we include a recent Talbach timeseries derived by linear interpolation from the Burgga discharge to avoid the average year. Hereby, we follow another reviewer's comment. We have added the number of years for which we have data in the text. The paragraph was changed accordingly:

We obtained 30 years of 15-minute discharge data from the Regional Council Freiburg for the Brugga River and from the State Institute for Environment Baden-Württemberg (LUBW) for the Dreisam. For the Talbach, only historical discharge data (daily resolution, 1980-2009, also from LUBW) were available. The historical Talbach discharge data showed a high correlation (R2 = 0.92) with the Brugga discharge. We therefore used Brugga discharges to obtain a recent Talbach discharge timeseries by linear interpolation. At Zipfeldobel, a flow meter (IFC 010 System, KROHNE, Duisburg, Germany) was installed in 2014 to log average discharge every 10 minutes. Flow meter data gaps (37% of all flow readings) were filled by linear interpolation after adding weekly manual discharge measurements to the data to bridge longer intervals with no data (see Fig. B1). This allows us to calculate annual discharge values in years with extended Flow-Meter downtime, thereby obtaining a more accurate value as described in the following section. Regarding the analysis of the timeseries, the flow meter was operational for most of the hydrological years 2023 and 2024; however, there is a data gap of two months in August and September 2024.

**Line 213-214 -> Over how many years was Mann-Kendall trend calculated?**

Because we removed the average year from the loads calculation due to another reviewer's comments, the calculation of Mann-Kendall now appears obsolete, as it was intended to strengthen the hypothesis that the average year is indeed representative.

While change in storage is often neglected in long term water balance, this term cannot be neglected at annual scale. Please add a stronger justification as to why this term was ignored for annual water balance. Were the groundwater levels at the start and end of time period same perhaps?

We agree with the reviewer's comment and now discuss storage changes in our discussion chapter 4.2. As suggested by the reviewer, we added a timeseries of groundwater levels in the Dreisam valley to the discussion, to support the hypothesis that storage conditions were equal at the beginning and end of the two-year sampling period.

Interpreting the two-year balance, we assumed that the storage state of the catchment was similar at the beginning and end of the sampling period. We justify this assumption by referring to groundwater levels close to the catchment outlet, which were similar at both times and close to the long-term mean (see Fig. 5).

Figure 1. Groundwater table at the DRC outlet (Ebnet, see Fig. 1) for the whole sampling period indicated by vertical grey dashed lines. The horizontal red line indicates the 60 year mean.

Furthermore, we now calculate a mass balance for both sampled years. Hydrological conditions were similar at the beginning and end of this period, as indicated by groundwater levels, allowing us to argue for a closed balance.

We calculated annual water in- and outputs for all catchments, covering the hydrological years 2023 and 2024, from November 1 to October 31, and for the whole sampling period.

Why was the water input into Zipfeldbel spring considered as precipitation occurring in nine pixels around the spring?

This approach aimed to obtain a more reliable estimate of precipitation at a specific location by reducing noise and interpolation artifacts. In this case, this does not make a huge difference (-25 mm for the 30 years mean precipitation if just a single pixel is used). We included this information into the draft:

To reduce noise and interpolation artifacts, we calculated the mean of nine cells surrounding the pixel at the Zipfeldobel spring.

**Also see the next comment on this:**

The aquifer feeding it could be receiving water from a broader area which could even be located far away. Therefore, the input should include precipitation over the recharge area. Has the recharge zone for the aquifer been delineated?

Mean catchment elevation is approximately 200m above the spring (estimated from a stable water isotope gradient in precipitation). Tracer studies showed that Tracer applied on the hillslope above the spring takes months to become visible in the spring water (Didszun (2000), https://www.hydrology.uni-freiburg.de/publika/FSH-BD19-Didszun.pdf). Because the spring is located on the side of a hillslope and not on the bottom of a valley, no clear catchment area can be derived from a DEM. Anyhow, because the theoretical size of the catchment area (~2 ha) is way below the precipitation interpolation grid size, and we assume uniform TFA concentrations, the exact location and shape of the catchment area are not important for our loads calculation.

We integrated this argumentation into the updated manuscript and stressed the resulting uncertainties:

This observation, however, is subject to uncertainties related to the interpolated Talbach discharge and potential errors in the estimation of the Zipfeldobel spring's catchment area. We emphasize that the catchment area of the spring is theoretical, as water may bypass beneath or near the spring. The actual recharge area cannot be inferred from topography due to the hillslope position and may change depending on the connection of underground flow paths, which vary according to season and wetness conditions. One should rather regard the contributing catchment of the Zipfeldobel spring as a theoretical ~2ha district within the 9 km² of the nine cells used for precipitation interpolation.

There could be subsurface flows occurring within the aquifer feeding the spring. If such flows are present, they should be accounted for in the water balance. Please clarify whether the

aquifer has known inflows or outflows or even other springs. If they are present but excluded, please provide a justification for neglecting them in the water balance.

The additional inflows and outflows of the springs aquifer can be excluded because, for the calculation of input and output masses, only a theoretical catchment area matters. We refer to the comment above for clarification.

What is the water transit time for the aquifer? What is the justification for considering the yearly change in storage term as zero? Possible reasons could be very long transit times, muti year stable groundwater levels or spring discharge, same groundwater level at the start and end of year, etc.

The exact transit times for the Dreisam aquifer have not been investigated. From groundwater tables and k-f values, we know that travel times in the order of several meters per day are possible. Therefore, transit times should be in the range of a few years, regarding a length of roughly 10 km for the Dreisam valley. Transit times at the springe are 2-3 years, according to Uhlenbrook et al. (2002) (Line 98). As mentioned above, we have now added a timeseries of groundwater values in the DRC-valley to show that conditions at the start and end of the sampling period were similar.

Line 260: Can you include the formula for calculating cdis and cGW? How is cQGW different from cGW?

We included the formulas in the draft:

$$\overline{c_{dis}} = \frac{\sum c_i Q_i}{\sum Q_i}$$

$$\overline{c_{GW}} = \frac{\sum c_i}{n}$$

cQGW is equal to cGW, we corrected the spelling mistakes.

**Results**

Line 276 -> It looks like the highest TFA levels in Dreisam river occur in May 2023. Also, it looks to me that precipitation was not that low during the winter. I suggest putting numbers to support these statements, since it is hard to judge them visually from the figure 2. Or perhaps

the intention was to highlight the high levels of TFA in river despite low levels of TFA in precipitation during winter. In that case, please rephrase the sentence accordingly.

Indeed, our intention was to highlight the high levels of TFA in the river despite low levels of TFA in precipitation during winter. We changed the sentence to:

Despite low TFA levels in precipitation, high TFA levels were observed in the Dreisam River during winter.

Line 306 -> Correlations for Deuterium excess is missing in Table 2.

We added deuterium excess to the table.

Line 315-> If 2023 as dry year and 2024 as wet year, then water balance for years 2023 or 2024 separately wouldn't be close without a change in storage term.

We added an interpretation regarding catchment storage to the discussion section 4.2. We replaced the average year balance with the two-year combined balance. We follow the previous suggestion of the reviewer and argue that similar groundwater tables at the beginning and end of the sampling period indicate a closed two-year balance. For details, see the comment regarding L 213-214 f.

Table 5 -> There is no equation 9 in the manuscript. Maybe it was accidently excluded.

This was a spelling mistake, as also mentioned by another reviewer. It should be Eq 8.

While it is reasonable to make assumptions like the WWTP TFA levels remained constant over a year, I still suggest explicitly stating somewhere that the TFA levels were based on a single day measurement.

We added the sentence to appendix E:

The calculation of total TFA load is based on a single-day measurement.

**And in the results section:**

The catchment's three WWTPs' TFA input, calculated from the single-day measurements, was minor (0.21 kg a-1; Table E1) and could be neglected.

While groundwater TFA levels seem constant over two years, observations over longer period might reveal a trend. Considering that anthropogenic sources have bene increasing, most probably the groundwater TFA levels have also been increasing here. Please explicitly

acknowledge this so that it doesn't give the impression that the groundwater TFA levels have stabilized.

We see that there is potential for misinterpretation. We followed the reviewer's advice and added a statement regarding trends in groundwater to the methods section.

Considering the relatively short sampling period, we emphasize that we cannot make any statements about long-term TFA trends in groundwater.

Considering that groundwater TFA levels could have been seasonal, was this accounted for incalculating mass flux?

Yes, we did not account for seasonal variation in groundwater TFA levels. We increased the uncertainty of the TFA value found for groundwater concentrations to 30% to reflect a potential seasonal trend in the results. Also see the comment on Line 149.

**Discussion**

The hypothesis on organic soil zone as temporal TFA storage which contributes to TFA pulses in streams during storm flows is interesting and insightful. This can be a good framework to explore TFA dynamics in other catchments as well. While the short-term mass balance suggests limited TFA retention, I would caution that a two-year dataset may not be sufficient to conclusively rule out retention processes. Given the known groundwater residence times in the Brugga catchment (ranging from a few to over ten years), some of the TFA currently reaching the stream could be originally from legacy sources, while recent TFA inputs may still be retained in subsurface. As such, I would suggest interpreting the apparent balance with some caution, and perhaps acknowledging the potential for longer-term storage and delayed transport within the system. I also suggest adding some discussion about systems with deep groundwater and long transit times where TFA might be retained more.

Acknowledging the uncertainties of an average year balance, we remove it from the draft. We now focus on interpreting the annual and the two-year mass balances. We thank the reviewer for the suggestions regarding additional discussion on legacy sources, long-term storage, delayed transport, and long groundwater transit times and the uncertainties they introduce to the apparent balance. However, we think that the influence of very old water from the deep crystalline groundwater systems is minor. We elaborate in an additional paragraph in the discussion section:

Further uncertainties in all catchments, as well as in the spring, may arise from legacy TFA stored within the hydrological system. Long transit times could influence transport processes, particularly along the extended flow paths within the crystalline bedrock. However, Richey et al. (1997) reported negligible retention in mineral soils, suggesting that crystalline bedrock does not serve as a significant reservoir for TFA. Moreover, historically lower TFA concentrations in rainfall would have resulted in lower concentrations within these systems. Therefore, although long transit times in the deep crystalline aquifer might delay TFA transport, the limited water flux, lower historical TFA inputs, and the weak binding affinity of TFA to crystalline rock collectively indicate that this effect is likely minor.

I appreciate the acknowledgement of potential uncertainties introduced by the use of data from a single precipitation sampling station as well as the spatial variability of TFA concentrations in precipitation. There is a TFA surplus in Dreisam catchment which you attribute to manures. However, could this surplus be from legacy storage of TFA from previous years? Given that 2023 was dry year, the TFA from previous years could have been retained in catchment and subsequently be mobilized in the wet year, 2024.

Also, due to another reviewer's comment, we estimated a possible input from liquid manure and found it to be negligible. We now state this in the updated manuscript. For the legacy storage, see also our answer above.

Assuming a TFA concentration of 100  $\mu$ g L-1 in liquid manure, as reported by the German Environment Agency (2023), an application rate of 15 tonnes per hectare (t ha-1) would result in an annual input of less than 1 kg of TFA across the entire agricultural area of the DRC (5.2 km2). Consequently, liquid manure can be considered a negligible source of TFA, taking into account the huge export surplus of over 200 kg a-1.

We shared the idea that TFA exported in 2024 could be legacy TFA from 2023. However, this should imply a lack of TFA export in 2023, which we could not observe. This opens a discussion on legacy TFA stored in soils in the form of precursor PPP and their degradation products, which we included into the discussion:

The increased export observed in 2024 was particularly pronounced in the DRC compared to the Talbach, Brugga, and Zipfeldobel catchments. The absence of an export deficit during the preceding dry year contradicts the assumption that the TFA export surplus in 2024 originated from the previous year. Since the surplus may stem from PPP use, discussing potential legacy storage is warranted. Indeed, PPP applications have been shown to leave residues in soils (Riedo et al. 2021). Moreover, prolonged transformation times in soils, dependent on moisture conditions, have been reported for PPPs such as Flufenacet, a compound that degrades into TFA (Scheurer et al. 2017; European Food

Safety Authority (EFSA) 2017). Therefore, the hypothesis that fluorinated compounds accumulate in soils, creating a potential TFA legacy storage, appears valid and is supported by our data.

The ET estimates for each year from water balance as well as from TFA mass balance rely on the assumption of closed water balance for each year. Therefore, I suggest including direct ET measurements, if they are available, or summarizing results of previous studies that report ET ranges for these catchments to support that these are reasonable estimates.

We appreciate the reviewer's suggestion and added a discussion on ET values in the literature.

Values for Brugga and Talbach (537 and 521 mm) were in the range of values reported in the literature. Didszun und Uhlenbrook (2008) reported 600 mm for the Brugga catchment, Wenninger et al. (2004) 530 mm for the Talbach basin, Hangen et al. (2001) 248 mm in winter, and 588 mm in summer for the nearby Conventwald study area, and Hoeg et al. (2000) 620 mm for the neighboring Zastler catchment.

TFA could be a valuable tool as tracer, especially considering that it is less expensive to measure than isotopes. Highlighting this could strengthen the case for its use as tracer.

We thank the reviewer for the suggestion. However, we are not quite sure if measuring TFA via LC-MS/MS is really cheaper than using a Cavity Ring-Down Spectroscopy system (e.g. Picarro L21x0-i) for water stable isotopes measurements. However, the direct injection method may improve in the future, and due to the recent surge in public attention, PFAS analytics in general could become more affordable. Because this is not the case yet, we did not include this point in the draft.

**Conclusion**

Line 470 -> This is a strong statement. While the hypothesis that organic soil zone is the primary TFA storage, and SSF is the primary mechanism by which it reaches stream has high chances of being true, we need additional data like soil TFA profile or isotope tracer studies to support this. Therefore, I advise you to rephrase this into a more cautious statement and acknowledge the need for direct measurements.

In general, I suggest a more cautious wording of conclusions to reflect the limitations and assumptions of the study. That said, these assumptions and limitations do not reduce the value of your work.

We thank the reviewer for the comment. We agree, framed the conclusions more carefully in general and reworked the specific section.

Our study suggests that the organic soil zone exhibits TFA storage, and SSF is the dominant process by which TFA is transported from soils to the river. We stress the need for direct measurements in soils confirming these observations.

**Minor comments:**

Line 54 -> "also" should be deleted.

We deleted the "also".

Line 56-57 -> "we took weekly sample of precipitation at a weather station and stream water in three nested catchments and a hillslope spring.

We rephrased the sentence according to the reviewers' suggestion.

Line 60 -> Consider changing to "headwaters, which are free of arable land."

We changed the sentence as suggested.

Line 84 -> This sentence is not clear and grammatically doesn't make sense.

**We changed the sentence to:**

The Ebnet waterworks, located in the lower Dreisam Valley, abstract 9 million m3 of groundwater per year to supply the city of Freiburg. Additional groundwater export from the aquifer below the Dreisam gauge was estimated at 60 mm (Didszun and Uhlenbrook, 2008).

Line 86 -> Move inclinations to earlier: "with inclinations up to 62°"

**We changed the phrasing:**

Here, 75% of the area is covered by steep, forested slopes with inclinations of up to 62°.

Figure 1 -> Consider making the background of labels on figure transparent or removing them from

figure, so that river network and catchment boundaries are visible.

We followed the reviewers' suggestion and replaced the white box with a transparent buffer:

Line 95-96 -> Since abbreviations "SOF" and "HOF" are not used even once after their introduction, please remove them

Because we added information on the proportion of HOF and SOF on discharge generation, we now need the abbreviations and use them further down in the text:

In terms of magnitude, SSF is the dominating process of discharge generation when compared to HOF and SOF (Steinbrich et al. 2016).

Line 121 -> For consistency, use "Fig. 1"

We now abbreviate the "figure".

... near the catchment outlet (Fig. 1).

Line 122 -> Consider using "released from" instead for better flow

We rephrased the sentence as suggested by the reviewer:

There, in addition to the ubiquitous atmospheric input, TFA can additionally be released from agricultural activities.

Line 128 -> The value of "n" does not add much meaning here and could be removed, unless you want to point out that the small number of samples makes this estimate uncertain. If so, it would be good to explicitly mention it.

We removed the value.

Concentrations of TFA in liquid manure and biogas digestate range between tens and hundreds of  $\mu g L^{-1}$  (German Environment Agency 2023).

Line 134 -> The full form of WWTP has already been introduced and therefore can be skipped here

We now abbreviate:

Scheurer et al. (2017) measured elevated TFA concentrations in the effluents of WWTP.

Line 142 -> Did you probably mean "river"?

The sentence is now phrased as follows:

We collected weekly streamflow samples from the river at a fixed position with turbulent flow.

Line 167-170 -> Consider enclosing A and B – abbreviations for eluents – in brackets or quotes or put them in italics to avoid confusion

We have now put them in italics.

Line 175 -> remove brackets around Synek, 2008

We removed the brackets.

Synek (2008)

Figure 2 -> Please correct the unit for Q in panel b. The green time series for spring looks like a solid line for most of the time except a short time period July-Oct 2024, when it is dotted line.

We changed the graphic accordingly.

Line 279 -> Consider specifying the months you mean by "late summer"?

The months will be included in the updated version.

At the end of drought periods in early November 2023 and October 2024, TFA concentrations in the Dreisam reached levels comparable to those in the Zipfeldobel Spring and the Brugga and Talbach Rivers.

Line 372 -> "On" should be used instead of "At".

We changed the section as the reviewer suggested.

Line 454 -> "Eq 9" is missing

It should have been equation 8, as mentioned above.

---

## Author Comment (AC3)

**EGUSPHERE-2025-2882, reply on RC3**

Comments are copied in plain text.

Replies are blue.

Changes to the manuscript are in green.

This study presents a valuable and comprehensive investigation into the atmospheric and terrestrial pathways, retention, and export of trifluoroacetate (TFA) at the catchment scale. The two-year dataset and multi-compartment sampling approach (precipitation, streamflow, springs, WWTPs) provide an important basis for understanding TFA dynamics. The manuscript is generally well written and organized; however, several methodological clarifications and interpretations should be addressed before publication.

We thank the reviewer for the time and effort that was put into the comments. We address the comments regarding methodology and other sections of our manuscript in the following answers.

Line 160–165:

Please clarify the rationale for choosing this separation column for TFA analysis. What are its advantages compared with other commonly used columns for PFAS analysis, such as the Hypersil Gold C18 column?

PFAS carry a hydrophobic fluorinated carbon chain, which would make them prone to binding to a C18-column. In the special case of the ultra-short chain TFA, this lipophilicity is reduced to a minimum (chain length is equal to one), and the negative charge on the organic acid group dominates the binding properties of the molecules to solid phases. Therefore, we use the AS17-C, which is optimized for separating anionic substances.

We changed the sentence accordingly:

The injection volume was set to 50  $\mu$ l, with separation performed on an IonPac AS17-C column (2 × 250 mm) and an IonPac AG17-C guard column (2 × 50 mm) (both from Thermo Fisher Scientific, Waltham, USA), which is optimized for separating small anionic molecules like TFA.

Line 165–170:

The role of 50 mM ammonium hydrogen carbonate in pure water as mobile phase A should be clarified. If the separation column is hydrophilic, the use of methanol as mobile phase B may not be ideal. Please confirm the column chemistry and justify the chosen mobile phase composition.

We confirm the use of a 50 mM ammonium hydrogen carbonate buffer and methanol. The combination of methanol and buffer is intended to modulate the polarity of the mobile phase to achieve divided loading and unloading of the column. Starting conditions with 20% buffer were sufficiently polar to carry TFA on the column from the aquatic samples. Increasing the buffer to 50% was enough to re-eluate all TFA from the column. We therefore chose this composition. We want to mention that the initial work on finding these gradients was performed by Scheurer et al. (2017).

We added a sentence for clarification:

... and all samples were analyzed using the following gradient: 20% of eluent A (0-1 min), 20-50% A (1-10 min), 50-20% A (11-16 min). This gradient was sufficient for efficient binding and elution of TFA.

**Line 175–176:**

You mention Mill-Q blank samples as procedural blanks. Were these blanks also used to assess potential TFA contamination from the LC-MS system (e.g., tubing, fittings, internal components)? Please describe any specific cleaning or pre-conditioning procedures used to minimize TFA background signals from the instrument.

The main way to ensure the absence of contamination from the LCMS was by measuring MQ samples before starting a batch. This usually took 1-3 samples until the background levels were stable below 20 ng/L, which we considered as negligible. We added the following sentence to account for the comment:

To prevent contamination from the LC-MS system, we measured MQ-blanks until the background levels were below 20 ng L-1 before each batch.

**Line 190-195:**

It is mentioned that the same separation and guard columns used for LC–MS analysis were also used for IC analysis of major anions and cations. Does this imply that the IC system could potentially be used for TFA determination as well? If so, was this tested or verified?

Columns for IC have a wider diameter (4 instead of 2 mm), but the column material is the same. Berger et al. (1997) measured TFA with IC during their spiking experiments, but LOD for IC with a conductivity reading was about 0.06 mg/L, which is about one to two orders of magnitude above environmental TFA concentrations. Hence, TFA measurement with IC is possible, but not for environmental concentrations.

Additionally, please clarify whether "supplier" refers to the supplier of the IC instrument or of the columns. Did you determine quantification limits for each ion using your own calibration curves under actual operating conditions, which might be more accurate than supplier-provided values?

The actual LOQ values are below the supplier-defined ones, but they differ for the different ions. We followed the reviewer's suggestion and included them in the text:

Actual LOQ values, calculated analog to the TFA LOQ, were below the supplier-defined ones (Na $^+$  0.83, K $^+$  0.18, Mg $^{2+}$  0.026, Ca $^{2+}$  0.15, Cl $^-$  0.66, NO $_3$  $^-$  0.126, SO $_4$  $^2$  0.188, all mg L $^{-1}$ ).

Actually, the suppliers for the column and the IC are the same. To avoid confusion, we specified this in the text as suggested by the reviewer:

Thermo Fisher Scientific reported a precision of 5% and an LOQ of 1 mg L-1 for the IC system.

Line 275–276:

Please elaborate on the factors leading to the highest TFA levels in the Dreisam River during the 2023–2024 winter.

We changed this section due to another reviewer's comment:

Despite low TFA levels in precipitation, high TFA levels were observed in the Dreisam River during winter.

We now elaborate on this in the discussion section.

We argue that elevated TFA concentrations in the main catchment were caused by the use of PPP on the arable land at the valley floor: The Dreisam exhibited higher ...

**Figure 2:**

Does the gray shading in panels (a–c) represent dry and wet conditions? Please specify this in the figure caption.

We apologize for the unclear phrasing and have changed the captions as the reviewer suggested:

Grey areas, showing changes from dry to wet conditions, are highlighted in Figure 3.

Line 303–304:

Based on the correlation data in Table 2, it seems that TFA showed positive correlations with all tracers except nitrate. Please confirm and revise accordingly.

We see the point that nitrate only correlates minimally with TFA in precipitation. We changed the section as the reviewer suggested:

In rainfall, TFA exhibited statistically significant positive correlations with all tracers except nitrate, showing the strongest associations with potassium and stable water isotopes.

**Line 306:**

The sentence "The same was true for the negative correlation with deuterium excess" is ambiguous. Please rephrase for clarity (e.g., "Similarly, TFA exhibited a negative correlation with deuterium excess").

We changed the sentence as the reviewer suggested.

A significant negative correlation was found for rain volume and deuterium excess (D-excess).

**Line 364–365:**

You state that "Potassium negatively correlated with TFA; however, concentrations were below LOQ and could not be reliably interpreted." How was the correlation established if potassium concentrations were not quantifiable? Please clarify or reconsider this statement.

The reviewer's observation is correct; this indeed sounds confusing. Anyhow, potassium could be quantified despite the concentration being below the supplier-defined LOQ. We now used the LOQ we found for our IC-system, and most  $K^+$  values were above this threshold, so we added an interpretation for potassium to the draft.

Potassium negatively correlated with TFA. Potassium in the spring water mainly stems from the weathering of silicate minerals in the soil or at the soil/rock interface. Therefore, low concentrations might indicate old water from deep aquifers, depleted of TFA. However, we advocate for care when interpreting the potassium-TFA correlation, because changes in potassium and TFA concentrations were minor and for potassium within the uncertainty range of the measurement (5%) (see Fig. F3).

Line 366–367:

Given that the spring pH is around 6, the previously cited finding that "TFA sorption to soils decreased with increasing pH up to pH 5" (Richey et al., 1997) may not adequately explain the observed behavior at the Zipfeldobel spring. Please discuss this limitation.

As the reviewer suggested, the findings on TFA sorption being dependent on pH do not apply to the spring because its pH is above 6. We reworded the paragraph:

Furthermore, pH showed a negative correlation with TFA. Prior findings indicated that TFA sorption to soils decreased with increasing pH. The increased sorption potential was observed up to pH 5 for soils with organic content smaller than 10% (Richey et al. 1997). The springs' primary flow path lies at the bedrock soil interface in the hillslope, where organic content is low. However, the spring's pH levels were above 6. Consequently, sorption and desorption are unlikely to be the driving processes explaining the correlation between TFA and pH at the Zipfeldobel spring.

**Line 428–430:**

Considering TFA's high mobility, the lack of significant retention in water bodies contrasts with reported TFA retention in plants and soils (Likens et al., 1997; Berger et al., 1997). Please elaborate on possible mechanisms or environmental conditions explaining this discrepancy.

There might be a misunderstanding. We think that TFA-retention from Likens and Berger et al. was primarily caused by their short sampling time. The mechanism that was missed was a delayed release of TFA from plants and soils.

We made it clear that the following sentences refer to the differences between the historical studies and the findings of our study. We rephrased the section:

...retention in plants and soils (Likens et al. 1997; Berger et al. 1997). Potentially, differences compared to our findings originate from the study design of both historic field experiments: Labeling with roughly a 1000-fold of today's annual background flux might have led to higher TFA uptake, and a sampling duration of less than one year might have missed the delayed release of previously taken up TFA. With up to 1 mg kg-1 dry weight (Freeling et al. 2022), leaves and needles might build up a substantial organic TFA pool. The decomposition of organic matter might release TFA from soils during wet conditions, aligning with the observation from the previous chapter (correlation with nitrate and discharge due to SSF through the soil zone). Both 1997-studies did not capture the wet winter following labeling in summer, and therefore might have missed delayed TFA export, resulting in higher retention values.

Line 430:

The sentence "Potentially, differences originate from the study design of both field experiments" should include more detail on what specific design differences (e.g., sampling frequency, soil types, hydrological setting) might explain the divergent results.

We added the fact, that both labelling studies form 1997 took place during summer and did not capture the winter export from litter TFA.

Please also see the comment above.

Line 447–449:

Please provide references supporting the statement that patterns/concentrations "were attributed to the distribution of TFA precursor molecules in the atmosphere."

The reviewer is right: little is known about small-scale variability of TFA in precipitation. We rephrased more carefully to show that this is speculative.

Furthermore, heterogeneities in the precipitation input might cause TFA export excess. Elevated concentrations in precipitation in the vicinity of cities (Wang et al. 2014; Freeling et al. 2020) might be attributed to the distribution of TFA precursor molecules in the atmosphere. Whether those observations hold on a meso-catchment scale remains unclear. The lower part of the DRC is located downwind of the city of Freiburg. Therefore, elevated precipitation concentration in the Dreisam valley might be possible. Consequently, the spatial variation of input concentrations near Freiburg might explain some differences in the agricultural excess TFA amounts.

Line 380 vs. Line 470:

You stated that "our hypothesis of a temporal TFA storage, most likely associated with organic soil, seems valid," yet later conclude that "We identified the organic soil zone as a primary TFA storage." Since the data suggest only temporary accumulation, the latter conclusion may overstate the findings. Please rephrase to maintain consistency and avoid overinterpretation.

We took the comment of the reviewer into account and rephrased more carefully:

The study suggests that the organic soil zone exhibits temporary TFA storage, and SSF may be the process by which TFA is transported from soils to the river.

**Line 475:**

You compare TFA loads from farming activities with values reported for precursor PPP degradation in Joerss et al. (2024). If those values were derived from a different catchment, the comparison may not be meaningful. Please clarify whether the data are directly comparable.

The quantification of agricultural TFA excess and the corresponding sections were removed from the draft following another reviewer's suggestion.